# Asynchronous glutamate release is enhanced in low release efficacy synapses and dispersed across the active zone

Philipe R. F. Mendonça[1,2 ✉], Erica Tagliatti[1], Helen Langley[1], Dimitrios Kotzadimitriou[1], Criseida G. Zamora-Chimal[3], Yulia Timofeeva [1,3 ✉] & Kirill E. Volynski [1 ✉]

The balance between fast synchronous and delayed asynchronous release of neurotransmitters has a major role in defining computational properties of neuronal synapses and regulation of neuronal network activity. However, how it is tuned at the single synapse level remains poorly understood. Here, using the fluorescent glutamate sensor SF-iGluSnFR, we image quantal vesicular release in tens to hundreds of individual synaptic outputs from single pyramidal cells with 4 millisecond temporal and 75 nm spatial resolution. We find that the ratio between synchronous and asynchronous synaptic vesicle exocytosis varies extensively among synapses supplied by the same axon, and that the synchronicity of release is reduced at low release probability synapses. We further demonstrate that asynchronous exocytosis sites are more widely distributed within the release area than synchronous sites. Together, our results reveal a universal relationship between the two major functional properties of synapses – the timing and the overall efficacy of neurotransmitter release.

[1] University College London Institute of Neurology, London, UK. [2] Department of Physiology and Biophysics, Federal University of Minas Gerais, Gerais, Brazil. [3] Department of Computer Science, University of Warwick, Coventry, UK. ✉email: p.mendonca@ucl.ac.uk; y.timofeeva@warwick.ac.uk; k.volynski@ucl.ac.uk

The synaptic transmission provides the basis for neuronal communication. When an action potential propagates through the axonal arbour, it activates voltage-gated $Ca^{2+}$ channels (VGCCs) located in the vicinity of release-ready synaptic vesicles docked at the presynaptic active zone[1]. $Ca^{2+}$ ions enter the presynaptic terminal and activate the vesicular $Ca^{2+}$ sensor Synaptotagmin 1 (Syt1, or its isoforms Syt2 and Syt9), thereby triggering exocytosis of synaptic vesicles filled with neurotransmitter molecules. The neurotransmitter diffuses across the synaptic cleft, binds to postsynaptic receptors and evokes further electrical or chemical signalling in the postsynaptic target cell. This whole process occurs on a timescale of a few milliseconds. Recent data demonstrate that such speed and precision are achieved in large part via the formation of nanocomplexes that include presynaptic VGCCs, vesicles belonging to a readily releasable pool (RRP) and postsynaptic neurotransmitter receptors[2,3].

In addition to fast, synchronous release, which keeps pace with action potentials, many synapses also exhibit delayed asynchronous release that persists for tens to hundreds of milliseconds[1,4]. The asynchronous release is potentiated during repetitive presynaptic firing and is triggered via activation of multiple sensors with both low (e.g. Syt1) and high (e.g. Syt7) $Ca^{2+}$ affinity[5,6]. Accumulating evidence demonstrates that the balance between synchronous and asynchronous release plays an important role in coordinating activity within neuronal networks, for example, by increasing the probability of postsynaptic cell firing and/or modulating action potential precision[7–10]. It is well established that asynchronous release levels vary among different types of neurons[1,10,11]. Interestingly, recent data show that the ratio between asynchronous and synchronous release can also be differentially regulated among presynaptic boutons supplied by the same axon and depends on the identity of the postsynaptic cell, which contributes to target cell-specific communication in the brain[7,8].

The mechanisms that control the relative contributions of synchronous and asynchronous release at the level of single synapses are, however, poorly understood. Variability in asynchronous release among different neuronal types has been attributed to differences in synaptic morphology (e.g. the coupling distance between RRP vesicles and VGCCs)[10] and to cell type-specific expression of components of the vesicular release machinery (e.g. the $Ca^{2+}$ sensors Syt1 and Syt7)[11]. Whether the same mechanisms contribute to the regulation of release modes among presynaptic boutons located on the same axon remains unclear. It is also unknown whether synchronous and asynchronous release occur from the same pools of synaptic vesicles. Recent functional electron microscopy analyses combined with fast high-pressure freezing have helped to visualise loci of either synchronous or asynchronous release events within a single active zone[12,13]. The application of this approach to synaptic populations indicates that synchronous and asynchronous release events tend to occur in different sub-domains of the active zone. However, how synchronous and asynchronous release sites are located with respect to each other within the same active zone has not yet been established.

To address these questions, in this work we developed a novel imaging technique and data analysis framework that allowed us to directly investigate the relationship between synchronous and asynchronous glutamate release, both at the level of individual active zones and across large populations of synaptic outputs from a single pyramidal neuron. We find that the balance between synchronous and asynchronous release varies widely among synaptic outputs of individual pyramidal neurons. We show that asynchronous release fraction is enhanced in synapses with low release probability and that asynchronous release sites are more scattered within the presynaptic release area than synchronous sites. Our findings are consistent with a model in which functional presynaptic properties are regulated by a synapse-specific adjustment of the active zone morphology and the effective coupling distance between presynaptic $Ca^{2+}$ channels and release-ready synaptic vesicles.

## Results

### Imaging of synchronous and asynchronous vesicular release across axonal arbour.
Our approach (Fig. 1) is based on the expression of the fluorescent glutamate reporter SF-iGluSnFR on the axonal membrane, which allows detection of glutamate release from individual synaptic vesicles with millisecond resolution[14–18]. We sparsely transfected neocortical neurons in culture with SF-iGluSnFR and established whole-cell voltage-clamp recording in a neuron with pyramidal morphology expressing the sensor (Fig. 1a). We next imaged action potential-evoked glutamate release by monitoring changes in SF-iGluSnFR fluorescence (at a rate of 4 ms/frame) in tens to hundreds of individual presynaptic boutons supplied by the axon of the selected neuron, in response to a 5 Hz train of 51 action potentials triggered by brief somatic voltage steps (Fig. 1b–e, Methods). By applying a set of spatio-temporal filters and calculating the maximal projection of the resulting image stack, we could identify all active boutons irrespective of their release probability, as long as they released at least one vesicle during the train (Fig. 1d and Supplementary Movie 1).

Evoked synaptic SF-iGluSnFR responses had a stereotypical waveform with a quasi-instantaneous rising phase that was followed by a slower exponential decay, corresponding to glutamate unbinding from SF-iGluSnFR molecules ($\tau \sim 68$ ms, Fig. 1e and Supplementary Fig. 1). The relatively slow decay rate of SF-iGluSnFR responses made it difficult to measure the amplitudes of release events that occurred in close succession (less than 50 ms) using the normalised SF-iGluSnFR fluorescence profile ($\Delta F/F_0$). Furthermore, we found that $\Delta F/F_0$ signals depended not only on the number of exocytosed vesicles, but also on the presynaptic bouton size and geometry, which vary significantly among synapses (Fig. 1e, Supplementary Fig. 5). Therefore, we applied a deconvolution procedure (using the averaged SF-iGluSnFR response waveform) to determine the precise timings and the amplitudes of release events (Fig. 1e, Supplementary Figs. 1–4 and Methods). The histograms of deconvolved event amplitudes at individual boutons typically showed between one and three discernible quantal peaks, consistent with the exocytosis of one or more vesicles of glutamate (Fig. 1e and Supplementary Figs. 5 and 6)[14,16,18]. By fitting the histogram with a sum of Gaussian functions, we estimated the amplitude of the SF-iGluSnFR signal corresponding to the release of a single quantum ($q$, Fig. 1e and Methods). Critically, the application of this quantal analysis allowed us to estimate the number of vesicles released during each event and therefore directly compare vesicular exocytosis among different synapses.

For each bouton, we calculated the total release efficacy, $n_T$, by dividing the sum of all quanta released during the train by the number of action potentials in the train ($N_{AP} = 51$). We note that $n_T$ is directly related to the overall release probability, $P_{rel}$. Indeed, let us consider a synapse with RRP size of $m$ vesicles and with an average release probability of individual RRP vesicles $p_v$. According to binomial statistics, the probability that at least one vesicle is released at a given action potential is $P_{rel} = 1 - (1 - p_v)^m$ and the total release efficacy (defined as the average number of vesicles released per action potential) is $n_T = mp_v$. In low release probability synapses, $n_T \approx P_{rel}$. However, in synapses

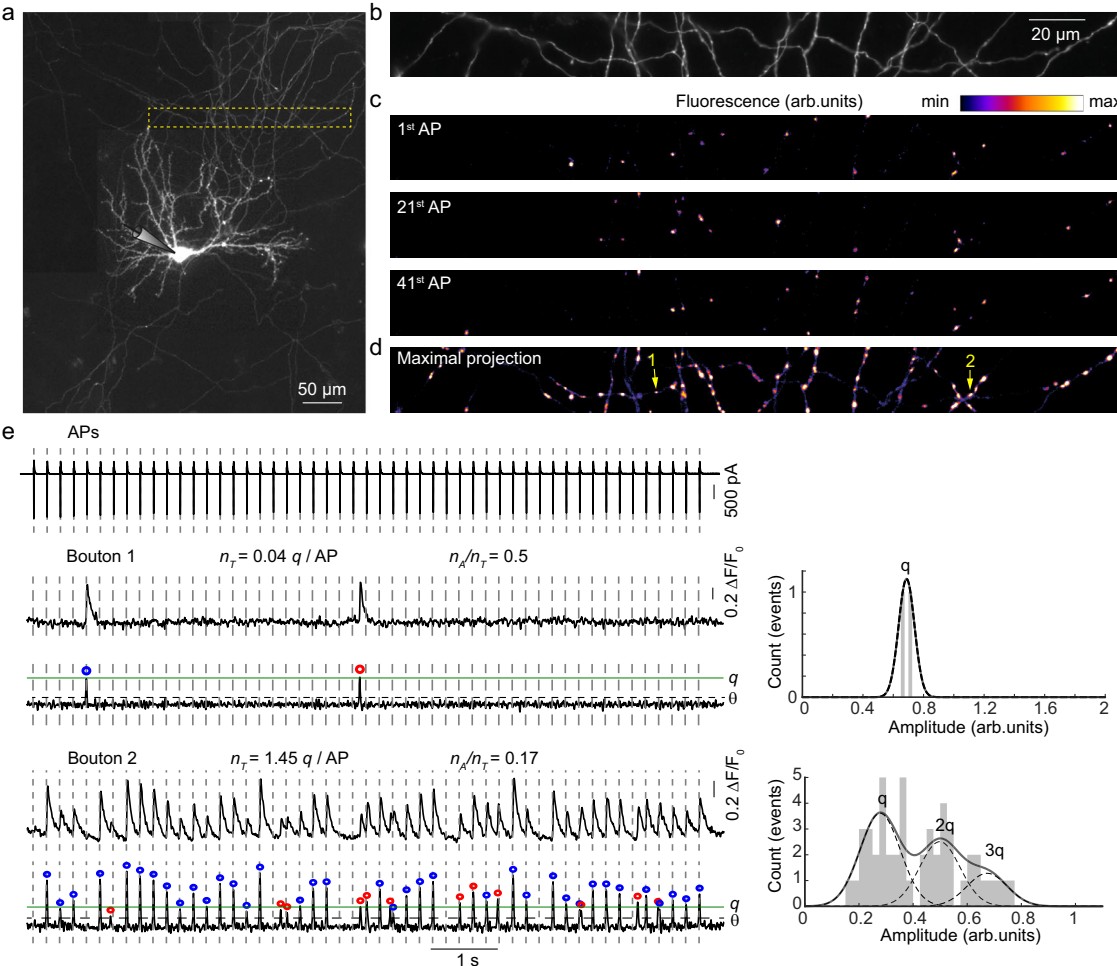

**Fig. 1 SF-iGluSnFR fluorescence imaging of quantal synchronous and asynchronous release in tens of individual presynaptic boutons supplied by a single axon.** A detailed illustration of the experimental paradigm in a representative wild-type neuron (total $N = 16$ neurons from 5 culture preparations were recorded). **a** Reconstructed mosaic image of a pyramidal neuron expressing the SF-iGluSnFR probe. **b** A region of interest (ROI) in the axonal arbour selected for imaging of presynaptic SF-iGluSnFR responses, corresponding to the yellow box in (**a**). **c** Heat maps of SF-iGluSnFR responses to the 1st, 21st, and 41st action potentials (APs) during a 5 Hz train of 51 action potentials. The images are averages of 3 frames from a band-pass filtered image stack immediately after each action potential (see Methods and Supplementary Movie 1). **d** Maximal projection of the band-pass filtered image stack, revealing locations of all presynaptic boutons that released at least one vesicle during the stimulation train. **e** Analysis of quantal SF-iGluSnFR responses in two representative boutons (Boutons 1 and 2 in **d**). Left, somatic action potential escape currents time-aligned with band-pass filtered and deconvolved SF-iGluSnFR signals. Quantal release events were identified as local maxima on the deconvolved traces located above the threshold $\theta$ (horizontal dashed lines) corresponding to 4 standard deviations of the background noise (see also Supplementary Figs. 2–6 and Methods). Blue circles mark synchronous and red circles mark asynchronous release events (see Fig. 2 for the definition of the synchronous/asynchronous release time threshold). Right, quantal analysis. To determine the amplitude of SF-iGluSnFR signal corresponding to release of a single vesicle ($q$, green lines on deconvolved traces) the positions of peaks on the amplitude histograms were fitted with a sum of 4 Gaussian functions (black lines) (Methods). This was then used to calculate in each bouton the total number of vesicles released per action potential (release efficacy $n_T$) and the fraction of asynchronous release events ($n_A/n_T$) (see Supplementary Figs. 5 and 6 for more examples).

that release on average more than one vesicle per action potential, the use of $P_{rel}$ is not optimal as it saturates at 1 (e.g. Bouton 2 in Fig. 1e). By contrast, $n_T$ provides a linear readout of vesicular release in all types of synapses. Therefore, we used $n_T$ in subsequent analysis. In full agreement with previous studies (e.g. refs. [16,19]) we found that $n_T$ varied widely among presynaptic boutons supplied by the same axon (~0.01–2.2 range, Supplementary Fig. 5 and also Figs. 3 and 4 below).

To define a time threshold for separation of synchronous and asynchronous release events, we calculated the average time course of vesicular exocytosis following an action potential across all recorded cells. This allowed us to compare the imaging data at single synapses to previous electrophysiological recordings from

synaptic populations. In agreement with the electrophysiological data (e.g. refs. [4,10,20–22]), the vesicular release time-course determined with SF-iGluSnFR followed a biphasic shape (Fig. 2a), with well-defined fast (<10 ms) and slow (>10 ms) release components. We note that although the majority of synchronous release is expected to occur within 1–2 ms after the presynaptic spike reaches the bouton[1], the onset of synchronous release is expected to be delayed by ~2–6 ms with respect to the somatic action potential, due to the finite speed of action potential propagation (as boutons are located several hundred micrometres away from the soma, Fig. 1a)[23]. This widens the distribution of detected synchronous release events and therefore the transition between the fast and slow-release components detected with SF-

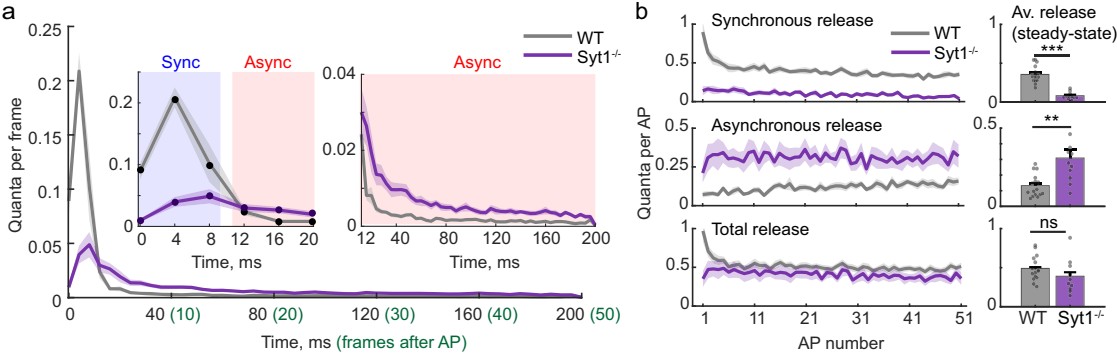

**Fig. 2 Comparison of vesicular release kinetics in wild type (WT) and Syt1$^{-/-}$ neurons. a** Average time courses of evoked vesicular release in WT and Syt1$^{-/-}$ (quanta per frame). The traces were obtained by first averaging in each imaged bouton the number of quanta detected in individual frames following each of 51 action potentials in the train (frames 0 to 50, 4 ms/frame). This was followed by calculating the cell-averaged responses (range 21 – 162 boutons per cell) and finally the mean responses across all recorded neurons (shaded areas represent SEM among the recoded cells). Based on the distinct biphasic time-course of vesicular release in wild-type neurons, the time threshold between synchronous and asynchronous release events was set at 10 ms after the somatic action potential (*i.e.* at the border between 2nd and 3rd frames, see main text for details). **b** Comparison of synchronous, asynchronous and total release in wild type and Syt1$^{-/-}$ neurons during the 5 Hz stimulation train. Left, traces represent the average number of vesicular quanta released at each action potential in a single presynaptic bouton (mean ± SEM, shaded area). Right, distributions of mean synchronous, asynchronous and total release efficacies at the steady state (between action potentials 20 and 51) among the recorded cells (mean ± SEM). (**a**, **b**) Wild type, $N = 16$ cells from 5 culture preparations; Syt1$^{-/-}$, $N = 11$ cells from 4 culture preparations ** $p < 0.01$, *** $p < 0.001$, ns $p = 0.09$, two-tailed Mann–Whitney U test (for exact $p$ values and for further details of statistical analysis see SourceData.xlsx file).

iGluSnFR occurs at ~10 ms, which was therefore used as a threshold for the separation of synchronous and asynchronous exocytosis components.

**Asynchronous release is elevated in synapses with low release efficacy.** Having established the protocol for imaging of different modes of vesicular release, we asked how synchronous and asynchronous events are distributed among synaptic outputs of individual pyramidal neurons. By using the defined 10 ms threshold, we classified events as synchronous or asynchronous (blue and red colour codes respectively in Fig. 1e and subsequent figures) and calculated in each bouton the efficacies of synchronous and asynchronous release ($n_S$ and $n_A$ respectively). By normalising these to the total release efficacy ($n_T$), we obtained the synchronous release fraction ($n_S/n_T$) and the asynchronous release fraction ($n_A/n_T$). As $n_S/n_T$ and $n_A/n_T$ are directly related ($n_S + n_A = n_T$ and $n_S/n_T = 1 - n_A/n_T$), in subsequent analysis, we refer solely to $n_A/n_T$.

We next sought to understand to what extent spontaneous vesicular release, which occurs in the absence of action potentials, contributes to the recorded evoked SF-iGluSnFR responses and, therefore, to our estimates of $n_A/n_T$. We clamped the membrane potential of the imaged cell at −70 mV and monitored quantal release in the absence of stimulation for 10 seconds, which corresponded to the duration of the 5 Hz train. We then recorded SF-iGluSnFR responses evoked by the 5 Hz train in the same set of ROIs. This allowed us to measure the frequency of spontaneous release in synapses that responded to action potential stimulation: 0.016 ± 0.004 Hz (Supplementary Fig. 7a–d). This value agrees with the estimate of spontaneous release frequency recently obtained using postsynaptic iGluSnFR imaging (~0.011 Hz)[24]. During the 10-second imaging window, we detected spontaneous events only in 13.3 ± 3.0 % of boutons. In contrast, we observed asynchronous events in 85.6 ± 1.7% of boutons (Supplementary Fig. 7e). Furthermore, the ratio between the number of spontaneously released vesicles (during 10 seconds interval) normalised to the total number of quanta released during 5 Hz train of 51 action potentials was ~0.8 %. This value is ~30 times lower than the asynchronous release fraction $n_A/n_T$ during 5 Hz stimulation: ~25% (Supplementary Fig. 7f and

Fig. 1). Thus, we concluded that spontaneous release does not significantly contribute to the asynchronous release recorded during 5 Hz train of action potentials.

In each recorded neuron, we observed vast variability in the asynchronous release fraction among the imaged boutons: $n_A/n_T$ ranged from 0 (no asynchronous release events detected) to 1 (exclusively asynchronous events detected) (Fig. 3a and Supplementary Figs. 5–8). In large part, this variability was due to limited sampling (especially in boutons with low release efficacy). For example, let us consider a bouton that on average releases only 2 vesicles during the stimulus train and has an expected value of $n_A/n_T = 0.5$. Then, during a single trial, there is a 25% chance of observing 2 synchronous events, a 25% chance of observing 2 asynchronous events and a 50% chance of observing 1 synchronous and 1 asynchronous event. Thus, estimates of $n_A/n_T$ and $n_T$ are prone to high error if only one trial is used. One way to overcome this difficulty was to record several sweeps from the same ROI and calculate the average $n_A/n_T$. However, due to the limitations caused by phototoxicity, this would restrict our recordings to a single ROI and would also significantly reduce the number of recorded boutons and the overall experimental success rate.

We therefore chose an alternative approach based on grouping boutons according to $n_T$ and calculating the average $n_A/n_T$ in each group. We plotted the moving average of $n_A/n_T$ versus $n_T$ in individual cells. Interestingly, we found that the asynchronous release fraction varied with the total release efficacy. We observed that boutons with low $n_T$ (below ~0.2) showed a consistently higher asynchronous release fraction than the average value of $n_A/n_T$ in the cell (Fig. 3a and Supplementary Fig. 8a). In line with this observation, we found a negative correlation between $n_T$ and $n_A/n_T$ in the vast majority of recorded cells (Supplementary Fig. 8a, panel ii). To quantify this phenomenon further, we performed Monte Carlo simulations of vesicular release in response to a 5 Hz train of 51 action potentials, in order to determine an optimal group size for averaging boutons (Supplementary Fig. 9). We found that in the case of low release efficacy synapses ($n_T$ range between 0.04 and 0.16, corresponding to the release of 2 to 8 vesicles per train), the averaging of at least ~7 boutons was required to obtain a meaningful estimate of the

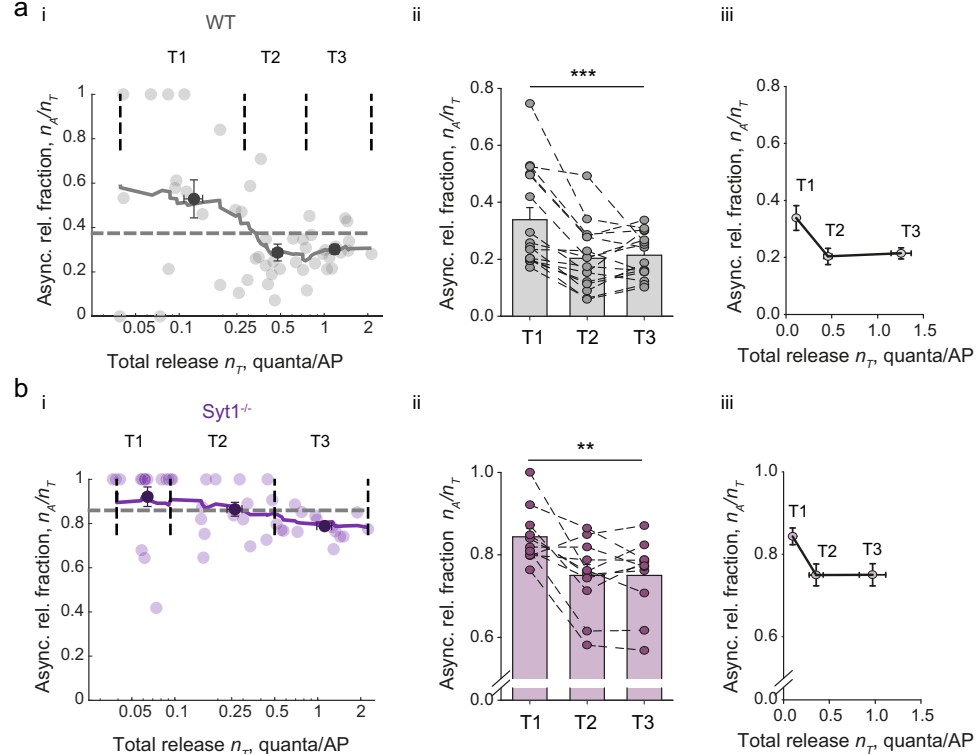

**Fig. 3 Asynchronous release fraction is increased in synapses with low release efficacy. a, b** Analysis of the heterogeneity in synchronous and asynchronous release among presynaptic outputs of individual wild type (WT) (**a**) and Syt1$^{-/-}$ (**b**) neurons during a 5 Hz train of 51 action potentials. (i) The relationship between asynchronous release fraction $n_A/n_T$ and the overall vesicular release efficacy $n_T$ among boutons in representative WT ($n = 50$ boutons) and Syt1$^{-/-}$ ($n = 46$ boutons) neurons (see also Supplementary Fig. 8 for more examples). Horizontal dashed lines depict the average fractions of asynchronous release in each cell. Continuous lines, moving averages (20 points span). Vertical dashed lines mark the subdivision of boutons into three terciles with low (T1), intermediate (T2) and high (T3) $n_T$ values. Black dots with error bars depict average values of $n_T$ and $n_A/n_T$ in each tercile (mean ± SEM). (ii) Bar and dot summary plots showing the increase in asynchronous release fraction in boutons with low $n_T$ (mean ± SEM), data from individual cells are connected by dashed lines. (iii) Dependency of $n_A/n_T$ versus $n_T$ averaged among all recorded cells (mean ± SEM). Data are from the same $N = 16$ wild type and N = 11 Syt1$^{-/-}$ neurons as in Fig. 2, each recorded neuron contained at least 21 boutons, ** $p < 0.01$, *** $p < 0.001$, repeated measures ANOVA (for exact $p$ values, all pairwise multiple comparisons, and further details of statistical analysis see SourceData.xlsx file).

$n_A/n_T$ value (80% confidence interval within ±0.15). As each recorded cell contained a minimum of 21 imaged boutons, we subdivided boutons in each neuron into three groups of equal size (terciles): T1 with low, T2 with intermediate, and T3 with high $n_T$ respectively. Indeed, we found an inverse relationship between overall release efficacy and the asynchronous release fraction: $n_A/n_T$ was ~1.6-fold higher in boutons with low release efficacy (T1, $n_T = 0.11 ± 0.01$ and $n_A/n_T = 0.34 ± 0.04$) when compared to boutons with intermediate (T2, $n_T = 0.46 ± 0.05$ and $n_A/n_T = 0.20 ± 0.03$) and high (T3, $n_T = 1.26 ± 0.11$ and $n_A/n_T = 0.21 ± 0.02$) release efficacies (Fig. 3a). The increase in the asynchronous release fraction in synapses with low $n_T$ was not reproduced in control Monte-Carlo simulations, where we randomly reassigned the type of each recorded release event (synchronous or asynchronous) in each bouton independently of $n_T$ using the cell-averaged $n_A/n_T$ value (Supplementary Fig. 10). This argues that the observed phenomenon was not a consequence of a spurious correlation.

It has been demonstrated that variability in the expression levels of the Ca$^{2+}$ sensor for synchronous vesicular release, Syt1, has a major role in the regulation of the balance between synchronous and asynchronous release among different types of neurons[11]. To test for a possible role of Syt1 in controlling the heterogeneity of vesicular release modes among synaptic outputs of a single neuron, we repeated our experiments in neuronal cultures from Syt1$^{-/-}$ mice. In line with previous reports[20],

deletion of Syt1 did not significantly change the overall release efficacy during the 5 Hz stimulation train (Fig. 2b, $n_T = 0.49 ± 0.04$ in wild type versus $n_T = 0.39 ± 0.06$ in Syt1$^{-/-}$) but resulted in a ~3 fold increase in the asynchronous release fraction ($n_A/n_T = 0.25 ± 0.03$ in wild type versus $n_A/n_T = 0.78 ± 0.02$ in Syt1$^{-/-}$). In spite of the overall increase of asynchronous release, we still observed the reciprocal relationship between the asynchronous release fraction and the overall release efficacy $n_T$ (Fig. 3b and Supplementary Fig. 8b). This indicates that the balance between different release modes at synapses supplied by a single neuron is at least in part determined by mechanisms distinct from the possible variation in the expression levels of Syt1.

**Relationship between short-term facilitation and synchronous/asynchronous release balance.** Typically, synapses with low release probability display short-term facilitation of vesicular release during repetitive stimulation[25,26]. Our finding that asynchronous release fraction is higher at synapses with low release efficacy synapses suggests that the balance between different release modes could be linked to the type of short-term plasticity expressed in the same presynaptic terminals. To test this hypothesis, we imaged vesicular release in response to 10 pairs of action potentials delivered at 20 Hz (50 ms inter-spike interval, 10 s between paired-pulse trials) and calculated in each recorded bouton the paired-pulse ratio (PPR), the asynchronous release fraction $n_A/n_T$ and the total release efficacy $n_T$ (Fig. 4a). We then

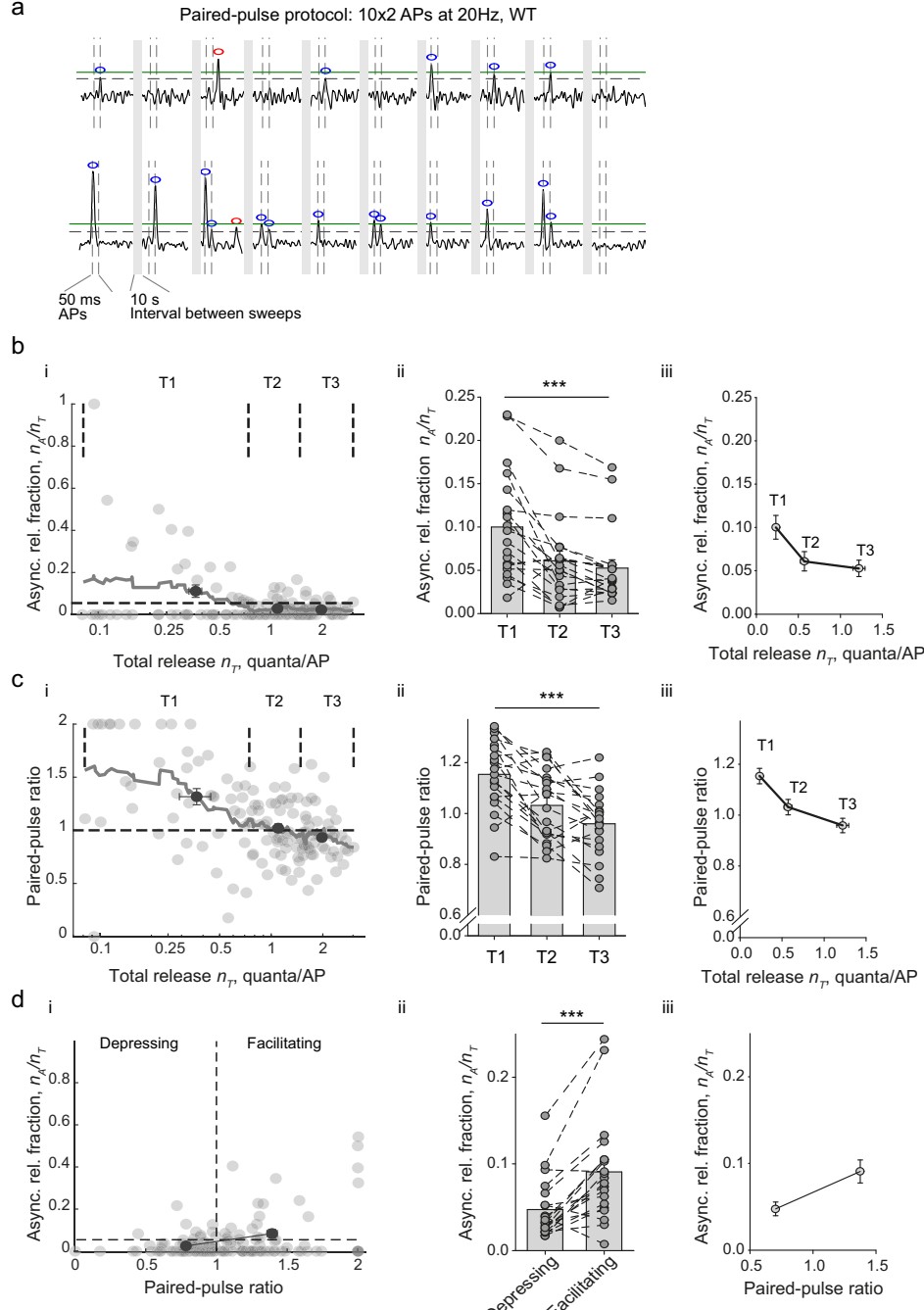

**Fig. 4 Asynchronous release is elevated in parallel with short-term facilitation in low release efficacy synapses. a** Analysis of the relationship between short-term plasticity, synchronicity and efficacy of glutamate release among synaptic outputs of single neurons. Paired-pulse stimulation paradigm and synaptic responses recorded in 2 representative boutons. As in Fig. 1, horizontal green lines show the amplitude of SF-iGluSnFR signal corresponding to the release of a single vesicle quanta ($q$). Vertical dashed lines depict action potential (AP) timings. Grey bars, 10 s intersweep intervals. Blue and red circles mark synchronous and asynchronous release events, respectively. **b–d** The relationships between: (**b**) asynchronous release fraction $n_A/n_T$ and the overall release efficacy $n_T$, (**c**) paired-pulse ratio (PPR) and $n_T$, and (**d**) $n_A/n_T$ and PPR in (i) a represenrtative cell ($n = 137$ boutons) and (ii, iii) across all recorded cells ($N = 20$ cells from 7 culture preparations, each recorded neuron contained at least 50 boutons). (i) Vertical dashed lines mark the subdivision of boutons either according to $n_T$ (T1, T2 and T3, as in Fig. 3) or according to PPR (Depressing/Facilitating). Horizontal dashed lines depict either the average fraction of asynchronous release or the average PPR. Black dots with error bars depict average values within each group (mean ± SEM). (ii) Bar and dot summary plots (mean ± SEM), data from individual cells are connected by dashed lines. (iii) Average values in each group among all recorded cells (mean ± SEM). *** $p < 0.001$, repeated measures ANOVA (**b** and **c**) and two-tailed paired $t$-test (**d**) (for exact $p$ values, all pairwise multiple comparisons, and further details of statistical analysis see SourceData.xlsx file).

compared the distributions of these major functional presynaptic properties among synaptic outputs of individual neurons (Fig. 4b–d).

We found that the overall asynchronous release fraction measured during paired-pulse stimulation (Fig. 4b) was lower (by ~3.5-fold) than that during the 5 Hz train (Figs. 2b and 3a). This result is in line with the well-established potentiation of asynchronous release during trains of action potentials, due to the accumulation of residual $Ca^{2+}$ (e.g. refs. [1,27]). Nevertheless, consistent with our findings with the 5 Hz train, the asynchronous release fraction measured during the paired-pulse protocol was also elevated (~1.8-fold) in boutons with low release efficacy (T1, $n_T = 0.23 \pm 0.02$ and $n_A/n_T = 0.10 \pm 0.01$) when compared to boutons with intermediate (T2, $n_T = 0.57 \pm 0.05$ and $n_A/n_T = 0.06 \pm 0.01$) and high (T3, $n_T = 1.22 \pm 0.07$ and $n_A/n_T = 0.05 \pm 0.01$) release efficacies (Fig. 4b).

In low release efficacy boutons, we often could not detect any responses to the first action potential among all 10 paired pulse trials, due to limited sampling. Therefore, to avoid division by zero, we adapted the classical expression for $PPR = N_2/N_1$ to $PPR = 2N_2/(N_1 + N_2)$, where $N_1$ and $N_2$ are the sums of vesicular quanta released at the first and at the second action potentials respectively. According to this definition, if a given bouton displays release only at the first action potential in the pair (i.e. $N_2 = 0$, maximal condition for short-term depression), then $PPR = 0$. Conversely, if release occurs only at the second action potential (i.e. $N_1 = 0$, maximum condition for short-term facilitation), then $PPR = 2$. If $N_1 = N_2$ (no facilitation or depression), then $PPR = 1$. In spite of vast variation of $PPR$ among individual synapses, averaging in individual cells revealed the expected inverse relationship between $n_T$ and $PPR$ (Fig. 4c).

Finally, we compared the relative distributions of $PPR$ and $n_A/n_T$ and found that the asynchronous release fraction was indeed ~2-fold higher in facilitating than in depressing synapses (Fig. 4d, $n_A/n_T = 0.09 \pm 0.01$ and $n_A/n_T = 0.04 \pm 0.01$ respectively).

**Super-resolution localisation of synchronous and asynchronous exocytosis sites.** We next asked, what are the spatial distributions of synchronous and asynchronous events within individual active zones? During vesicular exocytosis, glutamate can be assumed to be released from a point source. Therefore, the fusion of a single vesicle is expected to generate a bell-shaped SF-iGluSnFR fluorescence profile centred at the exocytosis site, which can be fitted using a 2D Gaussian function to determine the location of vesicular exocytosis with sub-pixel precision[17]. We applied this approach to directly compare the locations of synchronous and asynchronous release events within the same presynaptic bouton. By using quantal analysis, we first selected single-vesicle fusion events (Fig. 5a). We next generated the corresponding Event images for sub-pixel localisation of release sites, by applying pixel-by-pixel temporal deconvolution of the unitary SF-iGluSnFR response to the original image time series (Fig. 5b and Methods). The deconvolution procedure allowed us to specifically extract and amplify the spatial component of the SF-iGluSnFR fluorescence signal associated with the vesicular exocytosis event. This increased the signal-to-noise ratio and enabled us to localise positions of individual release events with approximately 75 nm precision (50–100 nm range, Fig. 5c, Supplementary Fig. 11 and Supplementary Movie 2). In 95% of boutons, vesicular exocytosis sites were clustered within a single compact area ($S = 0.09 \pm 0.01 \, \mu m^2$ in wild type and $S = 0.10 \pm 0.01 \, \mu m^2$ at Syt1 KO synapses, Fig. 4d, e and Supplementary Fig. 12), whilst the remaining 5% of boutons contained two separate release areas. We note that due to exclusion of multi-vesicular release events and single vesicle events that could not be localised with sufficient precision (i.e. within the specified 100 nm threshold, Methods and

Supplementary Fig. 11), we could only determine the locations of approximately 50% of all events. Furthermore, the imaged boutons were randomly tilted with respect to the microscope axes. Therefore, the release area determined with SF-iGluSnFR on the 2D image projection represents a lower limit of the corresponding active zone area[28]. Indeed, the average active zone size determined using 3D cryo-electron microscopy was ~1.35-fold larger (~0.12 $\mu m^2$) than obtained with our approach. The release area size varied widely among presynaptic boutons and, as expected, correlated with the total number of vesicles released per action potential ($n_T$) both in wild type and in $Syt1^{-/-}$ synapses (Fig. 5e).

**Asynchronous events are more widely distributed across the presynaptic release area.** To compare the relative locations of synchronous and asynchronous exocytosis sites, we applied hierarchical cluster analysis (Fig. 6a and Methods). In line with previous studies[2,28], the locations of individual release events could be grouped into small clusters (<100 nm diameter), which likely correspond to distinct release sites that are reused during repetitive stimulation. We found that among clusters that had at least two events, ~49% of clusters in wild type and ~43% of clusters in $Syt1^{-/-}$ neurons contained both synchronous and asynchronous events. This argues that the locations of synchronous and asynchronous events overlap and that they can occur from the same sites. However, in wild type neurons, the fraction of synchronous events progressively increased with the number of events in the cluster (Fig. 6a iii). Furthermore, the area covered by synchronous events had a shallower dependency on the number of events (slope 0.003 $\mu m^2$ / event) than the area covered by asynchronous events (slope 0.011 $\mu m^2$ / event) (Fig. 6b). Together, these findings suggest that synchronous release is confined to narrower nanodomains within the active zone. To test this at the level of individual synapses, we compared each bouton against reshuffled versions of itself (Fig. 6c, d). In each simulation, we maintained the number of synchronous and asynchronous events detected in each bouton but randomly scrambled their locations. We next calculated the average areas covered by each release mode (among all simulations) and used these to normalise the experimental values. Our rationale was as follows: a normalised value greater than 1 indicates that the real events have a sparser distribution than the random counterpart, whilst a normalised value smaller than 1 indicates that the real events have a more compact distribution. The reshuffling analysis showed that the normalised area was indeed greater for asynchronous than for synchronous release (Fig. 6d). We therefore conclude that asynchronous events in wild type synapses are more widely dispersed across the release area than synchronous events. By contrast, the overall prevalence of asynchronous release in $Syt1^{-/-}$ neurons largely occluded the difference between spatial distributions of synchronous and asynchronous exocytosis sites, although more compact clustering of the remaining synchronous release events was still detected with the re-shuffling analysis (Fig. 6d).

**Computational model of synchronous and asynchronous release.** Which mechanisms could account for the differential regulation of synchronous and asynchronous exocytosis at inter- and intrasynaptic levels? Our findings could potentially be explained by a model in which the observed variability of release properties results from structural differences in active zone organisation across different boutons, which lead to heterogeneity in the effective coupling distance between VGCCs and RRP vesicles. As the synchronous release is triggered by transient $Ca^{2+}$ nano/microdomains with a steep spatial gradient, the probability that a given RRP vesicle is released synchronously depends steeply on its coupling distance[29]. By contrast, the asynchronous

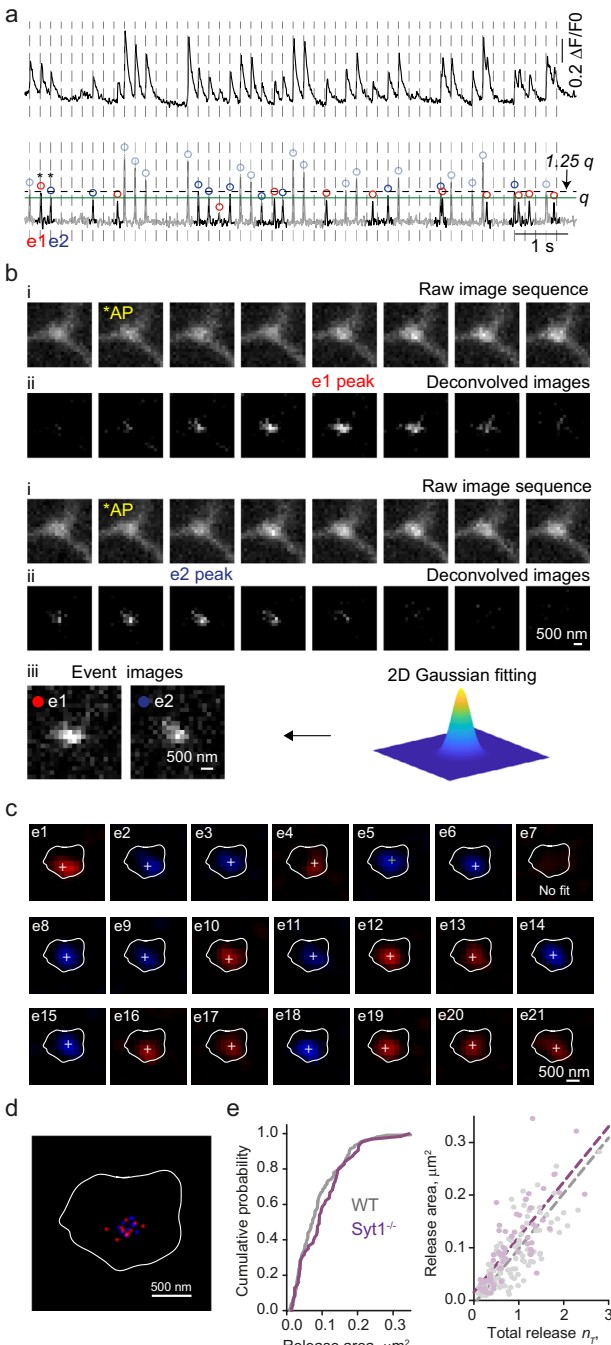

**Fig. 5 Super-resolution localisation of synchronous and asynchronous release events. a–d** Illustration of sub-pixel localisation analysis for single vesicle exocytosis events. **a** Raw (top) and deconvolved (bottom) SF-iGluSnFR traces from a representative bouton. Only single vesicle release events (black areas on the deconvolved trace) were included in the localisation analysis. These were selected using a 1.25 $q$ amplitude threshold (dashed line), where $q$ (green line) is the quantal size calculated as detailed in Fig. 1e. **b** Image pre-possessing in time domain. (i) Raw image sequences before and after event 1 (e1, asynchronous release, 3 frames after the preceding action potential, *AP) and event 2 (e2, synchronous release, 1 frame after the preceding action potential). (ii) Corresponding image sequences after pixel-by-pixel deconvolution of the unitary SF-iGluSnFR response. (iii) Event images, obtained by averaging 3 frames centred at the response peak on the deconvolved image stack used for subpixel localisation of release locations using 2D Gaussian fitting (Methods). **c** Sub-pixel localisation (white crosses) of single quantal events selected in (**a**) (see also Supplementary Movie 2). Images are after application of a wavelet filter (Methods); e7 could not be localised due to a low signal-to-noise ratio. The bouton outline (white line) was determined by calculating maximal projection of the deconvolved image stack. **d** Composite image showing relative locations of synchronous (blue) and asynchronous (red) release events within a compact release area. (**e**) Distribution of release area sizes and their dependency on the overall release efficacy in all boutons that passed the selection criteria for sub-pixel localisation (Methods, Data Inclusion and Exclusion Criteria) ($n = 106$ boutons from $N = 16$ cells for wild type and $n = 64$ boutons from $N = 11$ cells for Syt1$^{-/-}$ neurons).

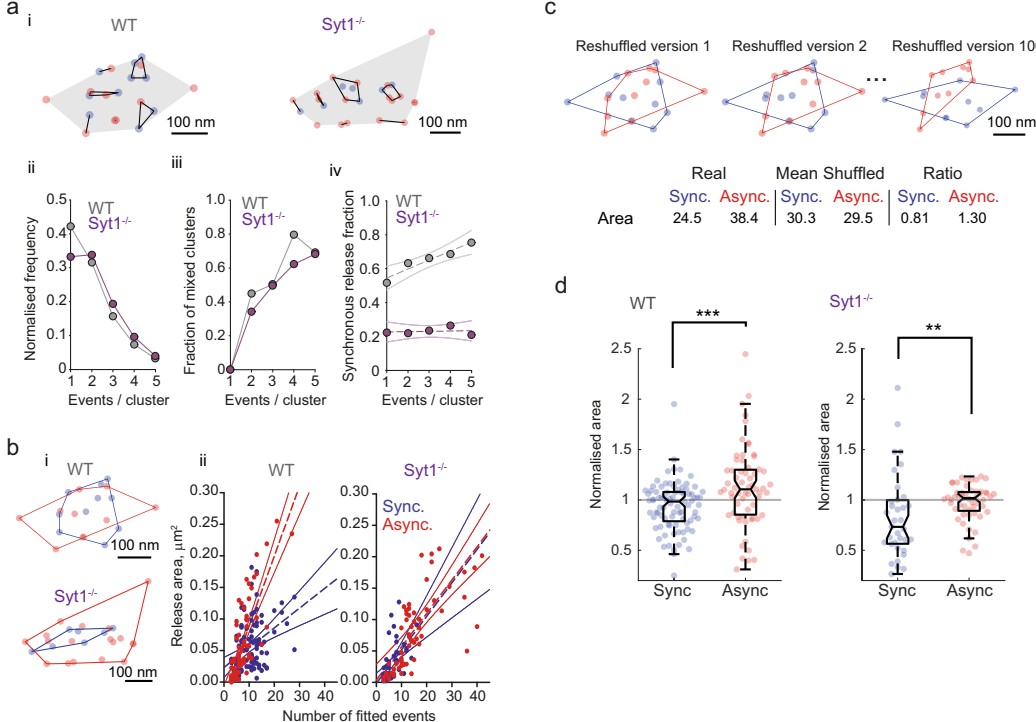

**Fig. 6 Relative distribution of synchronous and asynchronous release events in the active zone. a** Hierarchical cluster analysis of vesicular exocytosis sites. (i) Distributions of clustered events (clustering diameter threshold 100 nm, see Methods) in a representative wild type bouton (same as in Fig. 5a–d) and a Syt1$^{-/-}$ bouton. (ii) Distributions of number of events per cluster. (iii) Dependences of mixed clusters fraction (containing both synchronous and asynchronous events) on the number of events per cluster. (iv) Increase of synchronous events fraction with cluster size in wild type synapses. Dashed lines, linear regression. Solid lines 95% confidence intervals. (ii)–(iv) pooled data from $m = 799$ clusters from $n = 106$ boutons in wild type and $m = 553$ clusters from $n = 64$ boutons in Syt1$^{-/-}$ neurons. **b** Comparison of convex hull areas circumventing synchronous or asynchronous events. (i) Representative boutons (the same as shown in **a**). (ii) Relationship between the release areas and the number of fitted events in all analysed boutons. Dashed lines, linear regression. Solid lines, 95% confidence intervals. **c** Illustration of the reshuffling analysis used to compare spatial distributions of synchronous and asynchronous events within individual synapses (representative reshuffled versions of the wild type bouton shown in (**a**), see text for details). **d** Comparison of the normalised areas for synchronous and asynchronous events in wild type and Syt1$^{-/-}$ synapses. Data points and box-and-whisker plots: centre, median; notch 95% confidence interval for median; box, 25th–75th percentiles; whiskers, the most extreme data points not considered outliers (within 1.5 times the interquartile range from the bottom or the top of the box). ** $p < 0.01$, *** $p < 0.001$, two-tailed Mann–Whitney U test (for exact $p$ values see SourceData.xlsx file).

release is not expected to depend on the coupling distance, as it is triggered by a global ('residual') increase in the presynaptic Ca$^{2+}$ concentration after the collapse of Ca$^{2+}$ nano/microdomains (Fig. 7a)[1,30]. Therefore, it follows that an increase in coupling distance should lead (i) to a decrease of the overall release efficacy, with a concurrent increase in the asynchronous release fraction, and (ii) to a wider distribution of vesicular exocytosis within the active zone.

To test whether our results are consistent with this hypothesis, we explored the relationship between active zone morphology and the spatial distribution of synchronous and asynchronous release using experimentally constrained modelling of presynaptic Ca$^{2+}$ dynamics and activation of vesicular fusion.

The presynaptic bouton was modelled as a truncated sphere of radius $R_{bout} = 0.5$ μm with a single active zone located in the truncated surface of the sphere (Fig. 7b). Based on available ultrastructural data[12,31,32], the active zone was modelled as a circle with its area ranging from 0.03 μm$^2$ to 0.15 μm$^2$ (Fig. 7b, c). We assumed that VGCCs were located in small circular clusters of radius 30 nm and that each cluster contained 16 VGCCs (see Methods). We considered nine distinct cases of VGCC channel distribution in the active zone, which are illustrated in Fig. 7c. We varied the number of VGCC clusters (1–5 range) and also their relative distribution within the active zone. VGCC clusters were

either located more to the centre (Cases *i, ii, iv, v, vii* and *ix*) or to the periphery (Cases *iii, vi* and *viii*) of the active zone. In line with the experimental estimates, the overall channel density for all considered cases was ~ 500 μm$^{-2}$ (refs. [33–36]).

For each active zone geometry, we simulated the action potential-evoked Ca$^{2+}$ influx and three-dimensional buffered diffusion and extrusion using the Virtual Cell (VCell) computational platform[37,38] (Methods and our previous work[27,39–41]). In this way, we computed [Ca$^{2+}$] transients that occur at different locations in the active zone within the first 10 ms after an action potential. The obtained Ca$^{2+}$ dynamics were then used as an input for the allosteric model of the Ca$^{2+}$ activation of vesicle fusion[42] to calculate the probability of synchronous release ($p_s$) for individual RRP vesicles throughout the active zone (Fig. 7b, c). At present, due to the limited understanding of the roles of different Ca$^{2+}$ sensors in triggering of asynchronous release, there is no quantitative model of the Ca$^{2+}$ activation of asynchronous release in small excitatory synapses. Nevertheless, by assuming that the probability of asynchronous release ($p_a$) should not depend on vesicle location in the active zone (Fig. 7a), we constrained the $p_a$ value (between 0.02 and 0.03) based on the experimentally determined fraction of asynchronous release in recorded wild type synapses ($n_A/n_T \sim 0.25$, Fig. 2b and Supplementary Fig. 7f). In this way, we obtained maps for the

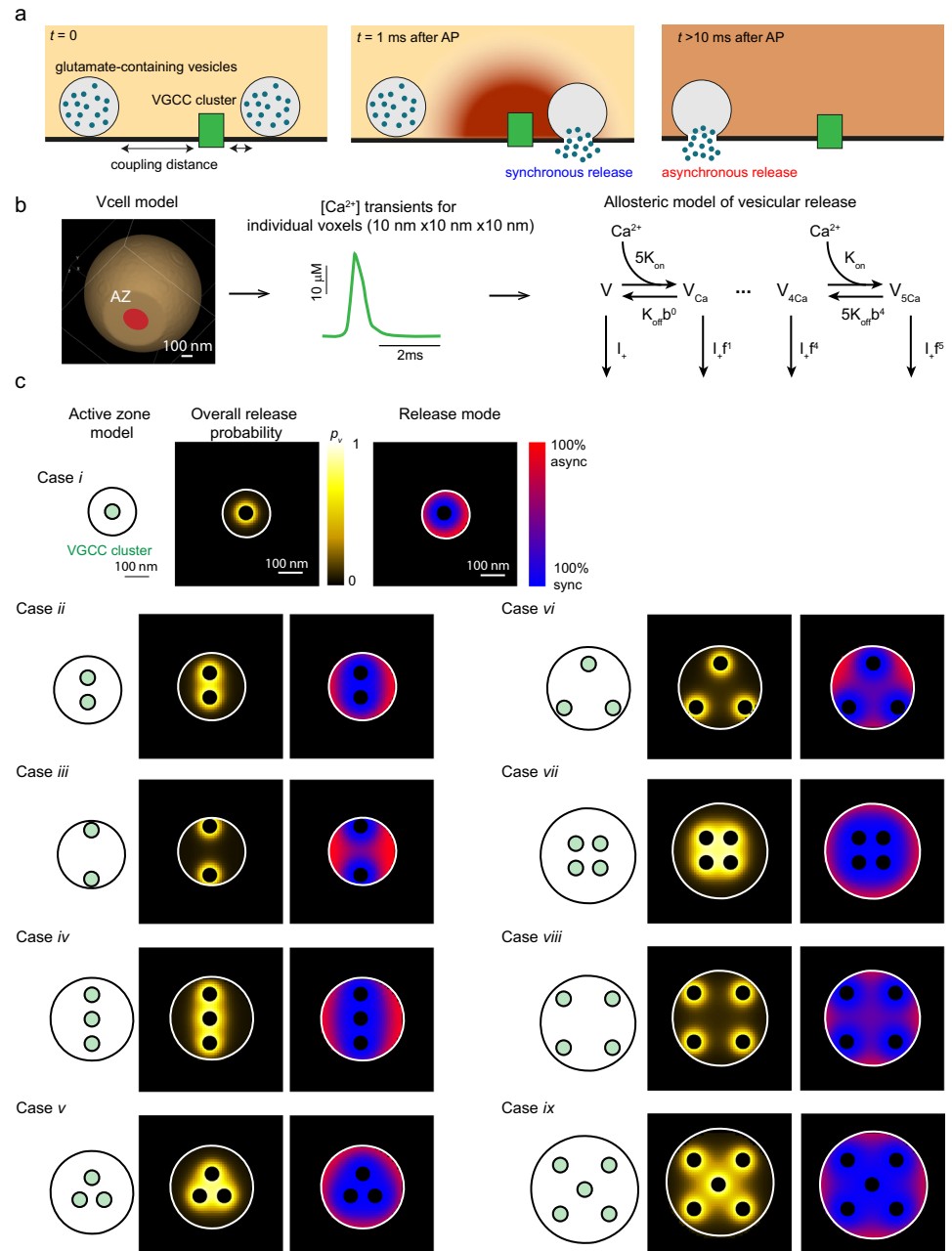

**Fig. 7 Modelling of synchronous and asynchronous release – generation of release probability maps for different active zone geometries. a** Schematics illustrating $Ca^{2+}$ triggering of different release modes. Synchronous release occurs within several milliseconds after an action potential and is triggered by transient local $Ca^{2+}$-nano/microdomains (~10–100 μM range) in the close vicinity of activated VGCCs (2–100 nm range). In contrast, asynchronous release occurs on a time scale of tens to hundreds of milliseconds and is triggered by longer-lasting global changes in presynaptic $Ca^{2+}$ concentration (~1-5 μM range)[1, 30]. **b** Schematics of the modelling framework. Action potential-evoked $Ca^{2+}$ dynamics in a presynaptic bouton (left) were simulated using the VCell computational environment on a 10 x 10 x 10 nm mesh (Methods). For each voxel located in the active zone, the obtained $Ca^{2+}$ transients (middle) were used as an input into the allosteric model of $Ca^{2+}$ activation of vesicular fusion[42] (right) to compute the probability of synchronous release for individual RRP vesicles $p_s$. As discussed in the text, the probability of asynchronous release $p_a$ was assumed not to depend on the vesicle location in the active zone and its value was constrained based on the experimental data (Methods). **c** Nine cases of considered active zone geometries (*i* to *ix*) and corresponding model-predicted maps of the overall release probability $p_v = p_s + (1 - p_s)p_a$, and of the relative fractions of synchronous (blue) and asynchronous (red) release modes for the model with endogenous buffering (see also Supplementary Fig. 13 for the model with the intracellular pipette buffer).

overall release probability of RRP vesicles $p_v = p_s + (1 - p_s)p_a$ and for the relative contribution of each release mode at different active zone locations (Fig. 7b). As the recorded cells were stimulated in the whole-cell patch-clamp mode (which leads to gradual replacement of the endogenous buffers with the pipette solution), we considered two limiting cases: a presynaptic bouton

with endogenous buffers (calmodulin, Calbindin D28k and ATP) (Fig. 7c) and a bouton filled with the intracellular pipette buffer (EGTA, ATP, K+-gluconate) (Supplementary Fig. 13) (Methods and Supplementary Tables 1 and 2).

We next randomly distributed RRP vesicles in the modelled active zone, assuming an average density of 100 vesicles per $μm^2$

(RRP size between 3 and 15 vesicles)[12,32] and simulated quantal release during a train of 51 action potentials using the obtained $p_v$ maps (Fig. 8 and Supplementary Fig. 14). For each active zone geometry considered, we computed the average $n_T$ and $n_A/n_T$ based on 250 simulations. As expected, $n_T$ increased with the active zone size, which was due in part to the concomitant increase in the number of RRP vesicles. Notably, the model also reproduced the experimentally observed inverse relationship between $n_A/n_T$ and $n_T$, showing that the asynchronous release fraction was lower in active zones of larger size. This was caused by an increase in the probability of synchronous release for individual RRP vesicles ($p_s$) in larger active zones, whilst ($p_a$) in our model did not depend on the active zone size and VGCC distribution (Fig. 7a, Fig. 8b–e and Supplementary Fig. 14b–e). The relative increase of $p_s$ in larger active zones was caused by the overlap of $Ca^{2+}$ nano/microdomains from multiple VGCC clusters, even though the overall density of VGCCs was the same in all modelled cases (Fig. 7c and Supplementary Fig. 13). Such VGCC cooperativity had maximal effect when VGCC clusters were located more to the centre of the active zone (e.g. compare Case *ii* versus Case *iii*, Case *v* versus Case *vi*, and Case *vii* versus Case *vii* in Figs. 7c and 8c).

We note that our assumption that $p_a$ does not vary with the active zone size is indeed supported by the experimental data. Structural analysis indicates that the average density of VGCCs in the active zone does not depend on the active zone size[33–36]. Because active zone area scales with the bouton volume[35], it follows that the global action potential-evoked increase in presynaptic $[Ca^{2+}]$, which triggers asynchronous release, should be on average similar in large and small presynaptic boutons. Indeed, several groups (including ours) have demonstrated using presynaptic $Ca^{2+}$ imaging that the amplitude of the normalised action potential-evoked $Ca^{2+}$ fluorescence response does not depend on the bouton volume[19].

To compare the model's output to our experimental findings, we processed the results of the simulations using the same analysis framework as for the experimental data set. To mimic the observed variability of release in individual cells, we combined the outputs from all nine considered geometry cases (10 simulations per case). In full agreement with the experimental results, the model predicted both the increase in the asynchronous release fraction in low release efficacy synapses (Fig. 8f and Supplementary Fig. 14f versus Fig. 3a) and that asynchronous release events are more widely distributed within the active zone than synchronous events (Fig. 8g and Supplementary Fig. 14g versus Fig. 6d). Finally, we compared the relationship between $n_A/n_T$ and $n_T$ for each considered active zone geometry. In all cases, we found the inverse relationship between $n_A/n_T$ and $n_T$, which was a consequence of the modelled variability in the average coupling distance between RRP vesicles and VGCCs across simulations (Supplementary Fig. 15).

## Discussion

The goal of this study was to understand the mechanisms that control the balance between synchronous and asynchronous release at the level of single presynaptic terminals. To address this question, we used the fluorescent glutamate reporter SF-GluSnFR and developed a method to image quantal vesicular release in tens to hundreds of individual synaptic outputs from single pyramidal cells in culture with 4 millisecond temporal and 75 nm spatial resolution.

Our results reveal a general principle that relates the two major functional properties of small central glutamatergic synapses – the probability and the synchronicity of vesicular release. We demonstrate that the ratio between synchronous and asynchronous synaptic vesicle exocytosis varies extensively among presynaptic boutons supplied by the same axon, and that the asynchronous release fraction is elevated in parallel with short-term facilitation at synapses with low release efficacy (i.e. low release probability).

Such fine-tuning of the balance between different release modes, together with the concurrent adjustment of release efficacy and short-term plasticity, has the potential to provide vast flexibility for synaptic computations in neural circuits composed of different cell types. Indeed, it is well established that vesicular release efficacy and short-term plasticity vary among synaptic outputs of cortical pyramidal cells according to the target cell type[43–46]. This allows for differential signalling via the same axon to different cell types, because facilitating and depressing synapses essentially act as high-pass and low-pass frequency filters respectively[25]. The link between the synchronicity of release, the overall release efficacy and short-term plasticity described in the present study thus suggests that the balance between different release modes is also likely to be tuned with the type of post-synaptic target cell. In line with this suggestion, two recent studies demonstrated cell target-specific regulation of the synchronous/asynchronous release balance in outputs of layer 5 pyramidal cells[7,8]. This hypothesis requires further systematic testing in different types of synapses.

Our analysis with super-resolution localisation of synchronous and asynchronous exocytosis sites shows that the locations of synchronous and asynchronous events overlap to a large extent. This argues that both release modes can occur from the same pool of vesicles. Furthermore, a wider distribution of asynchronous release loci indicates that the probability that a given vesicle will be released synchronously or asynchronously depends on its location within the active zone.

Two recent electron microscopy studies demonstrated that asynchronous release loci are more biased towards the centre of the active zone[12,13]. How can this finding be reconciled with the wider distribution of asynchronous release sites within the release area observed in the present work? There are several possible explanations for the observed discrepancy, which can be attributed to the differences in the methodologies used and their limitations. In the case of SF-iGluSnFR fluorescence imaging, by using 5 Hz stimulus trains, we can compare the locations of up to several tens of individual release events with respect to each other. However, this approach does not allow us to determine whether the detected events are located at the centre or the periphery of the active zone. In contrast, the 'zap-and-freeze' method used in the EM studies is based on high-pressure freezing of neurons at defined time points after a single action potential. This method allows one to localise positions of either synchronous of asynchronous release events within the active zone, but does not allow to determine the relative distributions of synchronous and asynchronous release events in the same bouton, as each active zone can be imaged only once at a selected time point.

Synchronous release events are expected to concentrate around VGCC clusters. One possibility is that VGCCs clusters are predominantly located at the periphery of the active zone. It then follows that asynchronous events, which are located further away from the VGCC clusters, can indeed be positioned closer to the active zone centre but be more widely distributed across the release area than synchronous events (see $p_v$ maps for Cases *ii*, *iv* and *viii* in Fig. 7c). Nonetheless, it should be noted that at present there is no direct data supporting the peripheral location of VGCC.

We also note that, due to technical limitations (Methods), we could only perform sub-pixel localisation of single vesicle release events. However, it is unlikely that the exclusion from our analysis of mostly synchronous multi-vesicular release events would

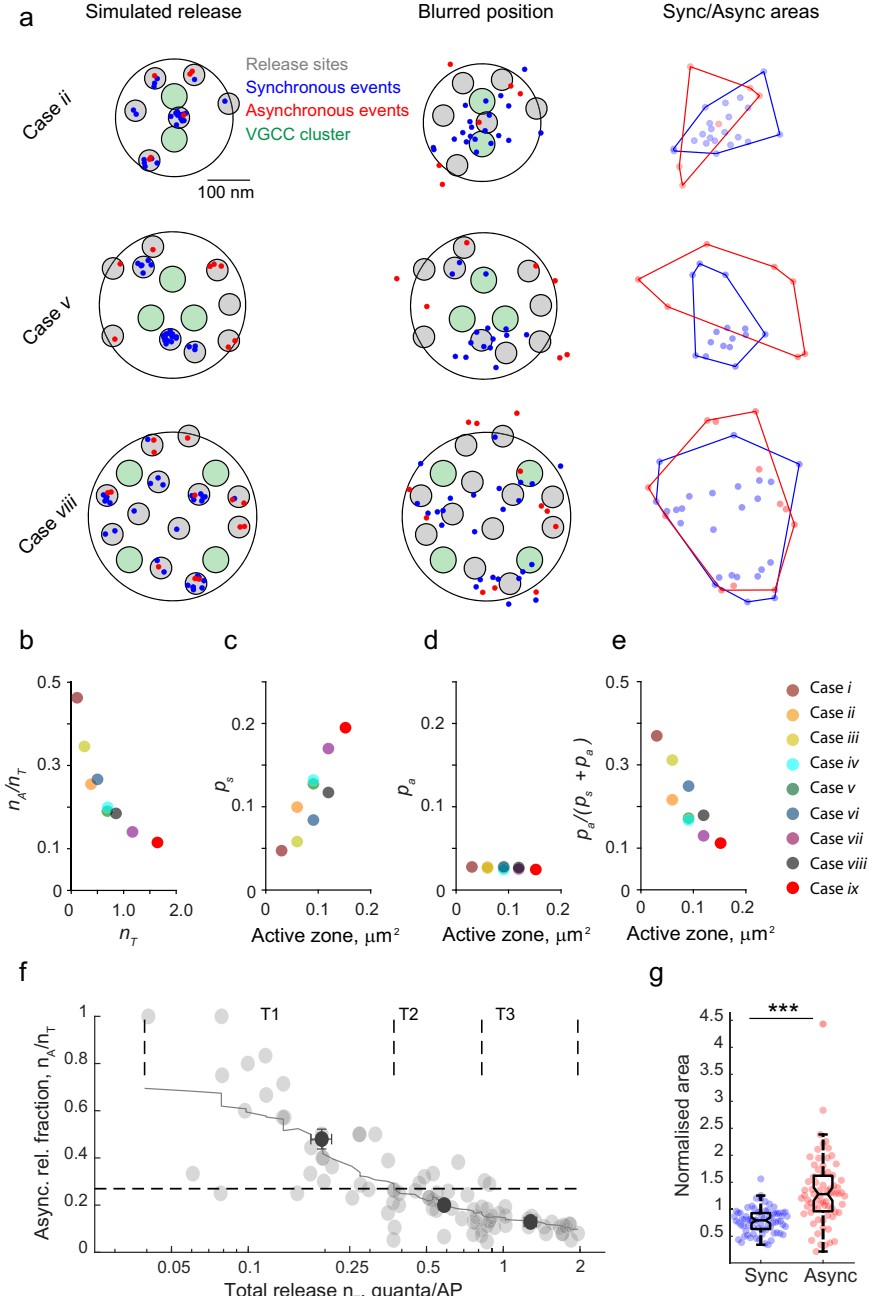

**Fig. 8 Simulations of vesicular release in synapses with different active zone geometries (model with endogenous buffers). a** Results of three representative simulations for active zone geometry cases *ii*, *v* and *viii*. Left, Simulated release. Grey circles depict randomly assigned locations of vesicular release sites. Blue and red dots correspond to synchronous and asynchronous release events respectively, which occurred during a simulated train of 51 action potentials. Middle, Blurred position. To mimic the experimental data, multi-vesicular release events were excluded and locations of the remaining events were blurred by adding a random value corresponding to the accuracy of SF-iGluSnFR event localisation (75 nm). Right, Sync/async areas. Comparison of convex hull areas circumventing simulated blurred locations of synchronous or asynchronous events. **b**–**e** Relationships between computed functional presynaptic parameters for nine modelled active zone geometries (averages of $n = 250$ simulations in each case): dependency of the asynchronous release fraction $n_A/n_T$ on the overall release efficacy $n_T$ (**b**) and dependencies of the probability of synchronous release $p_s$ (**c**), of asynchronous release $p_a$ (**d**) and of model-predicted asynchronous release fraction $p_a/(p_s + p_a)$ on active zone size, calculated for single RRP vesicles. Colour codes are shown on the right. **f**, **g** Comparison of the model's output to the experimental results. Combined outputs from all nine considered geometry cases ($n = 10$ simulations per case) were processed using the same analysis routine as for the experimental data set. **f** Model-predicted relationship between asynchronous release fraction and the overall vesicular release efficacy $n_T$. Similar to analysis in Fig. 3a horizontal dashed lines depict the average fraction of asynchronous release across all simulated boutons, continuous line – moving average, black dots – average values of $n_T$ and $n_A/n_T$ in each tercile (mean ± SEM). **g** Results of reshuffling analysis (similar to Fig. 6d). Data points and box-and-whisker plots: centre, median; notch 95% confidence interval for median; box, 25th–75th percentiles; whiskers, the most extreme data points not considered outliers (within 1.5 times the interquartile range from the bottom or the top of the box). ***$p < 0.001$, two-tailed Mann–Whitney U test (for exact $p$ values see SourceData.xlsx file).

change the observed wider distribution of asynchronous events. Indeed, it has been recently shown using vGlut1-pHluorin imaging that multi-vesicular release events overlap with uni-vesicular release events at sites closer to the centre of the release area[47]. Furthermore, even though the vGlut1-pHluorin localisation studies could not distinguish between synchronous and asynchronous events (due to the limited temporal resolution), Maschi and Klyachko[28] observed that evoked release events were more widely distributed within the same boutons following 10 Hz stimulation than following 1 Hz stimulation. As the 10 Hz stimulation protocol would be expected to evoke a larger asynchronous component of release, these results are consistent with our finding that asynchronous release events are more widely dispersed throughout the active zone.

We note that our experiments were performed at ambient temperature (23–25 °C). It has been demonstrated that increasing the temperature to near physiological levels (32–35 °C) leads to an enhanced synchronisation of vesicular release and a concomitant decrease in the asynchronous release fraction[21,22]. In large part, this is likely to be a consequence of temperature-dependent perturbation of the balance between the $Ca^{2+}$ signal amplitude within $Ca^{2+}$ microdomains in the active zone (which trigger synchronous release) and the global residual $Ca^{2+}$ signal (which triggers asynchronous release). Indeed, an increase in temperature is expected to affect the major determinants of presynaptic $Ca^{2+}$ dynamics, such as the action potential waveform, VGCC gating and $Ca^{2+}$ buffering and extrusion. Nevertheless, we note that due to the differences in the stimulation patterns, our experiments shown in Fig. 3 and Fig. 4 represent two contrasting cases with different ratios between the local $Ca^{2+}$ signal in the active zone and the global residual $Ca^{2+}$ signal. During repetitive stimulation (Fig. 3, 5 Hz train) the global $Ca^{2+}$ signal is expected to build up several fold in comparison to that after a pair of action potentials (Fig. 4)[27]. In line with this, the asynchronous release fraction was ~3.5-fold higher during 5 Hz trains ($n_A/n_T \sim 0.25$) than during paired-pulse recordings ($n_A/n_T \sim 0.07$). Still, in both cases, we observed an inverse relationship between $n_A/n_T$ and the overall release efficacy, $n_T$. This result therefore argues that the same relationship between release efficacy and $n_A/n_T$ is likely to be observed at physiological temperatures, despite the changes in presynaptic $Ca^{2+}$ dynamics and the overall synchronisation of release.

The systematic modelling of synchronous and asynchronous release in active zones of different sizes and containing different numbers of VGCC clusters, performed in this study, revealed several general principles that demonstrate how the active zone morphology determines the rates and distributions of synchronous and asynchronous release. The model predicts that the average probability of synchronous release of individual RPP vesicles ($p_s$) increases with the size of the active zone (Fig. 7c). This happens because, on average, larger active zones contain more VGCC clusters with overlapping $Ca^{2+}$ microdomains. Due to high $Ca^{2+}$ cooperativity of synchronous release, this $Ca^{2+}$ microdomain overlap leads to a significant increase in $p_s$ for RRP vesicles that are located in the vicinity of two or more VGCC clusters (Fig. 7c). By contrast, asynchronous release is triggered by the global residual increase in the presynaptic $Ca^{2+}$ concentration after the collapse of transient nano/microdomains (Fig. 7a). Therefore, the probability of asynchronous release ($p_a$) is not expected to directly depend on the distribution of VGCCs in the active zone.

The model reproduced both a greater asynchronous release fraction in low release efficacy boutons and a wider distribution of asynchronous release loci within the release area. This argues that the natural variability in the structural organisation of the active zone (including active zone size, the number of VGCC clusters

and their relative distribution with respect to RRP vesicles)[12,13,33–36] is the major factor that regulates the balance between synchronous and asynchronous release among synaptic outputs of pyramidal cells. Accumulating data demonstrate that the spatial organisation of VGCCs and RRP vesicles is linked to the distribution of AMPA and NMDA glutamate receptors in the postsynaptic cell[2,3,12,13]. It remains to be established whether and how these trans-synaptic interactions regulate the balance between different release modes.

Several recent studies have suggested that activity-dependent multi-step regulation of transient synaptic vesicle recruitment at the active zone is a major factor in determining efficacy, kinetics and plasticity of synaptic vesicle release at different types of synapses[13,31,48–50]. It remains to be established whether vesicular docking and priming states (and as a consequence, release site occupancy) are differentially regulated among presynaptic boutons supplied by the same axon. If this is indeed the case, this mechanism should also contribute to the interdependent regulation of vesicular release kinetics and short-term synaptic plasticity at a single bouton level. It also remains to be determined whether the balance between synchronous and asynchronous release in synapses supplied by the same axon is further tuned by the adjustment of the local abundance of different synaptotagmin isoforms.

## Methods

**Neuronal cultures and SF-iGluSnFR expression.** Experiments conformed to the Animals (Scientific Procedures) Act 1986, and were approved by the UK Home Office. Primary cortical neurons were produced from either wild type (C57BL/6J; RRID: IMSR_JAX:000664, purchased from Charles River) or Syt1$^{-/-}$ (B6; 129S-Syt1tm1Sud/J; RRID: IMSR_JAX:002478 The Jackson Laboratory) postnatal day 0 mouse pups of both sexes and cultured in Neurobasal A/B27-based medium (Thermo Fisher Scientific). The cortices were dissected and dissociated by enzymatic digestion in 0.25% trypsin for 10 min at 37 °C and then triturated using a standard p1000 micropipette. Neurons were plated on poly-L-lysine-treated 19-mm glass coverslips (1 mg/mL; Sigma-Aldrich) at a density of ~100,000 cells per coverslip placed in standard 12 well plates. At 5 days in vitro (5 DIV) neurons were transfected with pAAV.hSynap.SF-iGluSnFR.A184V plasmid[15] (addgene Plasmid #106174, 450 ng per coverslip) using Neuromag reagent (KC30800; OZ Biosciences). The transfection resulted in sparse expression of the iGluSnFR probe in a small subpopulation of neurons (~3%), which allowed us to select individual cells for imaging. Experiments were performed between 16 and 21 DIV.

### SF-iGluSnFR imaging of glutamate release

*Experimental set up.* SF-iGluSnFR fluorescence imaging experiments were performed on an inverted Olympus IX71 microscope equipped with a Prime95B 22MM back-illuminated CMOS camera (Teledyne Photometrics) using a 60x oil-immersion objective (1.35 NA) with resulting image pixel size 183.3 nm. SF-iGluSnFR fluorescence was recorded using a 470-nm excitation light-emitting diode (OptoLED Light Source, Cairn Research) and a 500–550 band-pass emission filter. Image acquisition was performed using µManager software[51]. Experiments were conducted in a custom-made open laminar flow perfusion chamber (volume 0.35 ml, perfusion rate ~1 ml/min) at 23–25 °C.

The imaging extracellular solution contained (in mM): 125 NaCl, 26 NHCO₃, 12 Glucose, 1.25 NaH$_2$PO$_4$, 2.5 KCl, 2 CaCl$_2$, 1.3 MgCl$_2$ (bubbled with 95% O$_2$ and 5% CO$_2$, pH 7.4). To ensure that recorded SF-iGluSnFR responses originate only from the stimulated axon, we suppressed recurrent activity in the neuronal network by blocking postsynaptic ionotropic glutamate and GABA receptors with (in µM) 50 DL-AP5 (Abcam), 10 NBQX (Abcam), and 50 Picrotoxin (Tocris Bioscience).

*Electrophysiology.* A putative pyramidal-like neuron expressing the SF-iGluSnFR probe that did not contain any other transfected cells in its vicinity was selected for imaging. A whole-cell voltage-clamp recording was established in the selected cell using a fire-polished borosilicate pipette (4-7 MΩ, Warner Instruments) and Axon Multiclamp 700B amplifier (Molecular Devices). The intracellular pipette solution contained (in mM): 105 K$^+$ Gluconate, 30 KCl, 10 HEPES, 10 Phosphocreatine-Na$_2$, 4 ATP-Mg, 0.3 GTP-NaH$_2$0, 1 EGTA (pH=7.3, balanced with KOH). The Multiclamp commander software was used to measure the series resistance (~20 MΩ), which was compensated at ~30%. Signal was acquired at 20 kHz (4-kHz Bessel-filtering) using a National Instrument board NI USB-6221 controlled with WinWCP software (created and provided by John Dempster, University of Strathclyde). A liquid junction potential of −10mV was subtracted from the measurements *post hoc*. The recorded neuron was held at −70 mV and action

potentials were evoked using 5 ms voltage steps from −70 mV to −10mV, producing stereotypical 'escape' currents.

*Fluorescence imaging.* After establishing a patch-clamp recording, evoked synaptic SF-iGluSnFR responses were imaged in 259 x 23 μm (1412 x 125 pixels) ROIs located within the axonal arbour, 200–1000 μm away from the soma. Images were acquired at 250 Hz sampling rate (4 ms/frame). The exposure time of each frame was captured with the data acquisition board. This allowed us to align the timing of captured frames to the electrophysiological recording, and therefore to determine the time between each vesicular release event and the preceding to action potential. For each neuron, between 1 to 4 different ROIs were imaged, with approximately 5 minutes interval between the trials. The stimulation protocols consisted either of 51 action potentials delivered at 5 Hz, or of 10 pairs of action potentials delivered at 20 Hz (50 ms inter-spike interval) with 10 seconds between individual paired-pulse trials.

**Image analysis.** Image analysis was performed offline using ImageJ (NIH)[52] and MATLAB (MathWorks) custom-developed scripts (Mendonca_et_al_AnalysisScripts.zip).

*Identification of active boutons.* In order to detect all active presynaptic boutons located within the chosen ROI, a series of filters was applied to the acquired image stack in the X-Y (space) and Z (time) dimensions. First, for each pixel, a moving average filter with a 3-point span was used to smooth the temporal profile of the SF-iGluSnFR responses. Next, a bandpass Gaussian filter (0.5 Hz–30 Hz) was applied in order to amplify the SF-iGluSnFR signal, revealing glutamate release sites by removing background fluorescence (Fig. 1c, d and Supplementary Movie S1). Next, a median filter (3x3 pixels) was used to reduce the spatial high-frequency noise component and to improve the robustness of the automatic detection of active boutons. Finally, a maximal projection of the filtered stack was obtained in order to visualise all regions where glutamate release events were detected, regardless of the rate of their occurrence. Positions of putative glutamate release sites on the maximal projection image were automatically detected using Find Maxima ImageJ plugin and a set of circular ROIs centred on the detected maxima (diameter 5 pixels, ~0.9 μm) was created. The detection threshold was chosen in a such way that all putative release sites were included, along with false-positive sites originating from background noise (these were excluded during quantal analysis stage as described below).

*Detection of quantal glutamate release events.* To determine the timings and the amplitudes of vesicular release events, the SF-iGluSnFR fluorescence signal from each selected ROI was first filtered using a 0.5 Hz – 30 Hz bandpass Gaussian filter, which allowed us to remove the baseline drift and the high-frequency noise. This was followed by deconvolution of the experimentally determined average unitary SF-iGluSnFR response, approximated as an instantaneous rise followed by an exponential decay $f_{iGluSnFR}^{unitary}(t) = e^{-t/\tau}$ with a decay time $\tau = 68$ ms (Supplementary Fig. 1): $f_{deconv}(t) = F^{-1}[F(f_{iGluSnFR}^{filtered}(t))/F(f_{iGluSnFR}^{unitary}(t))]$, where $F$ is the discrete Fourier transform and $F^{-1}$ is the inverse Fourier transform functions (MATLAB). The obtained deconvolved trace was further filtered using a 0.5 Hz – 30 Hz bandpass Gaussian filter to improve signal-to-noise ratio. In line with previous reports[18,53], the all-point histogram of the deconvolved trace could be well-approximated by a single Gaussian centred at zero: $\frac{A}{\sigma_{BN}\sqrt{2\pi}}\exp(\frac{-x^2}{2\sigma_{BN}^2})$ (Supplementary Fig. 2a). The obtained standard deviation $\sigma_{BN}$ (characterising the level of baseline noise) was then used to set the bouton-specific threshold for the detection of quantal events at $\theta = 4\sigma_{BN}$[53]. The amplitudes and the timings of individual release events were then determined at local maxima on the filtered deconvolved trace above the threshold.

*Quantal analysis.* To determine the amplitude of deconvolved SF-iGluSnFR signal corresponding to release of a single quanta ($q$) and the distribution of quantal vesicular release events in each bouton, we generated a quasi-continuous amplitude histogram using a bootstrapping procedure with added bouton specific noise ($m= 100,000$ simulations). The simulated values were obtained by randomly selecting an experimentally determined amplitude and adding a random number from a Gaussian distribution with a standard deviation $\sigma_{BN}$. The positions of quantal peaks on the obtained histogram were fitted using a finite mixture model consisting of the sum of 4 Gaussians: $\sum_{i=1}^{4}\frac{A_i}{\sigma\sqrt{2\pi}}\exp\left(\frac{-(x-\mu_i)^2}{2\sigma^2}\right)$, where $A_i$ is the amplitude of the $i$th peak, $\mu_i = \sum_{k=1}^{i}\lambda^{(k-1)}\mu_1$ is the average SF-iGluSnFR event amplitude corresponding to simultaneous release of $i$ vesicles, $\lambda$ is a factor that accounts for possible progressive saturation of SF-iGluSnFR signals during multi-vesicular release events, $\sigma^2 = \sigma_{BN}^2 + \sigma_{AN}^2$ is the variance of each peak, and $\sigma_{AN}^2$ is the noise component associated with variability of SF-iGluSnFR amplitudes (including the variability caused by a random jitter between the timings of release events and the camera exposure cycle, Supplementary Figs. 3 and 4). The use of bootstrapping allowed us to eliminate the error associated with the sensitivity of fitting procedure to the histogram bin size. The mean value of the saturation factor $\lambda$ estimated across $n = 1130$ boutons from 16 WT neurons was 0.9 (coefficient of variation

0.15). Thus, in line with a previous study in ribbon synapses[18], the amplitudes of deconvolved SF-iGluSnFR signals provided a nearly linear read-out for multi-vesicular release.

*Temporal resolution and sensitivity of quantal analysis.* To estimate the temporal resolution and the sensitivity of SF-iGluSnFR quantal analysis we simulated a set of SF-iGluSnFR synaptic responses ($n = 2000$) using the experimentally determined signal-to-noise ratio (Supplementary Fig. 2b) and processed the obtained traces using the filtering and deconvolution procedures described above. Analysis of simulated traces verified that temporal resolution of detection of vesicular release events was primarily limited by the camera acquisition rate (4 ms) (Supplementary Fig. 4a). The simulations confirmed that the use of $\theta = 4\sigma_{BN}$ detection threshold effectively abolished the presence of false-positive events (5 false positive in 1000 simulated traces). However, the fraction of false-negative (missed) events was increased with the decrease of signal-to-noise ratio (Supplementary Fig. 4c). We therefore excluded boutons with signal-to-noise ratio below $5\sigma_{BN}$ (~ 17% of all boutons, which are likely to be boutons outside the focal plane), thus limiting the fraction of false-negative events to ~20%.

*Sub-pixel localisation of vesicular release sites.* For each active bouton, a 21 x 21-pixel ROI (3.85 x 3.85 μm), centred at the corresponding intensity maximum on the maximal projection filtered stack, was extracted from the raw SF-iGluSnFR image stack. If the distance between two neighbouring boutons was less than 16 pixels (~3 μm), they were excluded from the analysis.

To increase the signal-to-noise ratio, the extracted image stacks were pre-processed pixel-by-pixel in the time domain. The traces were processed using a bandpass Gaussian filter (0.5 Hz–30 Hz), which was followed by deconvolution of the unitary SF-iGluSnFR response from the filtered signal, as described in the Identification of active boutons section. A vesicular release event was considered to contain a single quantum if its amplitude was below a 1.25q threshold (determined using quantal analysis). For each single quantal release, an Event image was calculated by averaging 3 frames from the deconvolved image stack centred at the response peak (Fig. 5b). The event image was then used for sub-pixel localisation of the vesicular release site with the ThunderSTORM ImageJ plugin[54].

The analysis consisted of two steps: finding the approximate position of the release site and its subsequent sub-pixel localisation. The event image was filtered using a wavelet filter, with a B-spline function of order 3 and a scale of 2. The approximate position of the release site was determined by finding a local maximum on the filtered image (intensity is greater than the specified threshold and at the same time greater than or equal to the intensities of 8 neighbouring pixels). Because the signal-to-noise ratio varied among synapses, the optimal threshold value was set independently for each bouton. For this purpose, we generated a Background noise image stack consisted of 1000 images. Each Background noise image was obtained by averaging 3 randomly selected frames from the deconvolved image stack, which were separated from the nearest release event by at least 15 frames (or 60 ms). The optimal bouton-specific threshold was then determined by running the analysis iteratively on the Background noise image stack, starting from a high level of intensity and decreasing it by half at every iteration until some events were detected. A second iterative process increased the threshold value in 250 intensity level steps, until no maxima were detected. The determined threshold value was next used to analyse the event images from the same bouton. Event images where the approximate position of the release site could not be determined with the selected threshold were excluded from the analysis. Because we preselected images corresponding to release of single vesicles, we a priori knew that there should be only one spatial maximum in each image, therefore we excluded images if more than one local maximum was detected. Finally, the sub-pixel localisation of single quanta release events that passed all the above selection criteria was performed by fitting an integrated form of 2D Gaussian function into the 7 x 7 pixel array centred around the established approximate positions[54].

*Estimation of precision for release site localisation.* The precision of localisation of vesicular release sites depends on several key factors. First, as in the case of single-molecule localisation microscopy, the localisation accuracy depends on the spatial resolution (point spread function, PSF) of the microscope and on the strength of SF-iGluSnFR signal (*i.e.* the signal-to-noise ratio determined by the number of photons collected for a given release event). In addition to this, the localisation accuracy of release events also depends on the spatial profile of the activated SF-iGluSnFR molecules within a given active zone and on the relative position and tilt of the imaged active zone with respect to the microscope focal plane[28].

To account for the joint effect of all the above factors, we estimated the localisation accuracy using an empirical approach. For each event we generated an Added noise image stack (in total 50 images). Each of the images in the Added noise stack was a sum of the original event image and a randomly selected image from the Background noise image stack described in the Sub-pixel localisation of vesicular release sites section above. We next applied the localisation analysis to individual images from the 'added noise' stack and calculated the average position of all fits (Supplementary Fig. 11b). The distance $\delta$ between the initial fit and the average position obtained from the Added noise image stack was used as an empirical estimate of the localisation accuracy (Supplementary Fig. 11c, d) and events with $\delta$ above the 100 nm threshold were excluded from analysis.

To test if $\delta$ represents a realistic estimate of the localisation accuracy, we performed similar analysis using artificial computer-simulated images. The rationale was that in this case we a priori knew the true locations of release sites. The simulated Event images were obtained using the following steps. We first simulated a spatio-temporal SF-iGluSnFR profiles corresponding to single vesicle fusion events using a general function form:

$$F(x,y,t) = F_{BG}(x,y) + \Delta F \exp\left(\frac{-(x-x_0)^2 - (y-y_0)^2}{2\sigma_{xy}}\right) \exp\left(\frac{-(t-t_0)^2}{\tau}\right),$$

where $(x_0, y_0)$ and $t_0$ are coordinates and timing of a release event; $\tau = 68$ ms, SF-iGluSnFR decay rate (Supplementary Fig. 1), $\sigma_{xy}$, spatial width of SF-iGluSnFR response (randomly selected from the experimentally determined range 250-450 nm), $F_{BG}(x,y)$ and $\Delta F$ background SF-iGluSnFR fluorescence and the amplitude of responses respectively (randomly selected from the recoded boutons). The obtained $F(x,y,t)$ responses were mapped on the experimental spatio-temporal grid (183.3 nm in space and a 4 ms in time domain) and noise was added using a randomly selected value of signal-to-noise ratio from the experimental data set (Supplementary Fig. 2b). Finally, the locations of simulated vesicular release events were fitted using the same routine as for the experimental data. The analysis of artificial images verified that $\delta$ provides an accurate estimate of the precision of localisation (50–100 nm range) (Supplementary Fig 11g). Furthermore, the computer simulations demonstrated that the use of the averaged position determined from the Added noise stack provides a more reliable estimate for the location of release site. Therefore, we used this improved averaged fit in the analysis of the relative distributions of synchronous and asynchronous release events in Fig. 6.

*Hierarchical cluster analysis.* Hierarchical cluster analysis was used to investigate the distribution of synchronous and asynchronous events among different release sites (Fig. 6a and Supplementary Fig. 12b). The built-in MATLAB functions `linkage` and `cluster` were used to best group events based on a distance threshold, allowing individual release cites to be defined by events located within 100 nm of the cluster centroid (in accordance with the precision of localisation of individual exocytosis events). The same algorithm was applied to identify synapses with two active zones (see examples in Supplementary Fig. 12). If boutons displayed event clusters with centroids distanced by 700 nm or more, they were defined as having two active zones and were excluded from spatial analysis.

**Computational modelling of synchronous and asynchronous release within the active zone.** The modelling of three-dimensional action potential-evoked presynaptic Ca²⁺ dynamics and of Ca²⁺ activation of vesicular release was performed using the presynaptic terminal model implemented in the Virtual Cell (VCell) simulation environment using the fully implicit finite volume regular grid solver and a 10 nm mesh (http://vcell.org)[37,38], and custom-developed MATLAB (MathWorks) scripts as described in detail in our previous work[39–41]. Briefly, the presynaptic bouton was considered as a truncated sphere of radius $R_{bout} = 0.5$ μm (described by the equation $[x^2 + y^2 + z^2 \le 0.25][z \le 0.42]$, all distances are in μm). The kinetic reaction rates for the presynaptic endogenous and intracellular pipette exogenous buffers were previously reported in refs. [55–62] and are specified in Supplementary Table 1 and Supplementary Table 2 respectively. Active zone sizes and VGCC cluster distributions are specified in Fig. 7. Based on the immunogold electron microscopy analysis (Fig. 8 in ref. [35], see also refs. [33,34]) and on our previous work, where we determined the relative contributions of different VGCC subtypes to action potential-evoked presynaptic Ca²⁺ influx in small glutamatergic synapses in culture (Fig. 5f in ref. [41]), we considered that each VGCC cluster contained 7 P/Q-type, 8 N-type, and 1 R-type VGCCs. Ca²⁺ extrusion by the bouton surface pumps (excluding the active zone) was approximated by a first-order reaction: $j_{extr} = -k_{extr}([Ca^{2+}] - [Ca^{2+}]_{rest})$, with $k_{extr} = 640$ μm s⁻¹ and $[Ca^{2+}]_{rest} = 50$ nM[41,55]. To compute the release probability map of synchronous release ($p_s$), we used the phenomenological allosteric model of the Ca²⁺ activation of vesicle fusion (Fig. 7b)[42]. As discussed in the results section, we assumed that the probability of asynchronous release was uniform across the active zone with $p_a = 0.03$ for the case of endogenous buffers and $p_a = 0.02$ for the case of intracellular pipette buffer (to match the experimentally determined fraction of $n_A/n_T \sim 0.25$).

Synchronous release of a given RRP vesicle was simulated by considering the overall release probability as the product $p_s \cdot p_{oc}$, where $p_{oc}$ is the probability that a given release site is occupied by a RRP vesicle. Because vesicular release during 5 Hz stimulation was on average depressed by ~ 2 fold in comparison to the response at the 1ˢᵗ action potential (Fig. 2b), we considered $p_{oc} = 0.5$. Asynchronous release was simulated only in the case of synchronous release failure.

**Data Inclusion and Exclusion Criteria.** In addition to the exclusion criteria described in the above sections, we applied the following selection criteria. In Fig. 3a, b, boutons with at least 2 release events were selected to allow meaningful estimation of $n_A/n_T$ and each recorded cell contained at least 21 boutons. Similarly, in Fig. 4, boutons with at least 2 release events were selected and each recorded cell contained at least 50 boutons. Previous work[28], where the vGlut1-pHluorin probe was used for sub-pixel localisation of vesicular release sites, demonstrated that positions of release events can be reliably fitted only within ~ 100 nm from the

imaging plane in z-direction and that selection of boutons with at least 5 detected events effectively limits the analysis to a sub-population of active zones that are mostly parallel to the image plane (within a 20º tilt). We therefore applied the same selection criteria in our analysis in Figs. 5 and 6. Furthermore, in order to compare the relative locations of synchronous and asynchronous release events (Fig. 6), only boutons that contained both event types were included.

**Statistical analysis.** The distribution of data in each set of experiments was first tested for normality using the Kolmogorov Smirnov test. For normally distributed data two-tailed Student's t-test for group means, paired t-test or One-Way Repeated Measures ANOVA with post hoc all pairwise comparison using Holm-Sidak (two-tailed) method were used as indicated. For data sets that failed the normality test two-tailed Mann-Whitney U test was used as indicated. The detailed statistical analysis is presented in SourceData.xlsx file. All data are presented as mean ± s.e.m. unless specified otherwise. Where possible, all data points were shown on the plots. No statistical methods were used to predetermine sample sizes, but our sample sizes were similar to those reported in previous publications that use similar techniques[14,17,18,41]. Data analysis was performed blind to the conditions and genotype tested. All statistical tests were performed using SigmaPlot 11 (Systat Software) and MATLAB (MathWorks) software packages.

**Reporting summary.** Further information on research design is available in the Nature Research Reporting Summary linked to this article.

## Data availability
Source experimental and modelling data are provided with this paper (SourceData.xlsx file). Raw images are available upon reasonable request from the corresponding authors. Source data are provided with this paper.

## Code availability
Custom MATLAB and ImageJ codes are provided with the paper within Mendonca_et_al_AnalysisScripts.zip

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

## Acknowledgements

We are grateful to Drs Shyam Krishnakumar, Dimitri Kullmann, James Rothman and Christopher Kushmerick for reading the manuscript and providing critical feedback. We thank Dr Hugh Robinson for kindly providing the Gaussian filter bandpass MATLAB function used in our imaging and trace processing. This work was supported by: The Wellcome Trust Strategic Award 104033/z/14/z (K.E.V.); Epilepsy Research UK Project Grant P1806 (K.E.V.); The Wellcome Trust PhD Studentship 203795/Z/16/Z (H.L. and K.E.V.); Medical Research Council UK Project Grant MR/T002786/1 (Y.T. and K.E.V.); The Virtual Cell is supported by NIH Grant R24 GM137787 from the National Institute for General Medical Sciences.

## Author contributions

Conceptualisation: P.R.F.M., K.E.V. Performing experiments: P.R.F.M., E.T., H.L., D.K. Development of data analysis framework: P.R.F.M., C.G.Z.C., Y.T., K.E.V. Computational modelling: Y.T., K.E.V. Writing – original draft: P.R.F.M., Y.T., K.E.V. Writing – review and editing: P.R.F.M., E.T., H.L., D.K., C.G.Z.C., Y.T., K.E.V. Funding acquisition: H.L., Y.T., K.E.V. Project supervision: K.E.V.

## Competing interests
The authors declare no competing interests.
