## [Peer review file · Nature Communications]

REVIEWER COMMENTS

Reviewer #1 (Remarks to the Author):

It is generally accepted that fast, synchronous neurotransmitter release requires synaptic vesicles primed close (within tens on nm) to clusters of voltage-gated calcium channels. This is because the calcium sensor for synchronous release, synaptotagmin 1, requires a high calcium concentration to be activated, and this is only present very close to open calcium channels and for the first few ms after an action potential. Until recently, this model was based solid but indirect inference. Recent studies using live imaging of release and localization microscopy have given a more direct picture: vesicle fusions tend to cluster within between one to a few sites within the active zone, corresponding to clusters of proteins comprising the release machinery.

Asynchronous release, by contrast, can be triggered by lower, 'residual' amounts of calcium. Based on this, one could make two related predictions: 1) a vesicle could undergo asynchronous fusion across a broader distribution within the active zone, not just close to calcium channel clusters since residual calcium fills the bouton, and 2) larger vesicle-channel coupling distance and a consequently lower release probability should bias a synapse more towards asynchronous release since vesicles closer to calcium channels are expected to have a higher (synchronous) release probability, with no comparative effect on asynchronous release.

Testing these predictions requires that live-cell localization of release events within the active zone be extended to asynchronous release. Using the glutamate sensor iGluSnFR and a sophisticated image analysis pipeline, Mendonça et al. both perform quantal analysis on a synapse- by-synapse, neuron-by-neuron basis, and localize synchronous and asynchronous release within the active zone. They find:

1) Synapses with lower release probability and, relatedly, higher paired-pulse ratios have a higher ratio of asynchronous/synchronous release. This is independent of Syt1, which mediates synchronous and suppresses asynchronous release, as the correlation remains in syt1 knockout neurons. This also holds on a per-release site basis: more active release sites have higher synchronous/asynchronous ratios.

2) Synchronous release and asynchronous release both cluster around the same apparent release sites, but asynchronous release tends to occur over a broader area. Simulations that place calcium channels at the periphery of the active zone, consistent with recent findings, can account for this distribution, as well as the previous finding that asynchronous release is biased towards the center of the active zone.

This paper is a substantial advance, providing the first experimental evidence for a broader distribution of asynchronous that has been predicted previously, and fits these findings into a coherent intellectual framework that accounts for the spatial organization of synchronous and asynchronous release. I recommend it for publication, with the following changes requested:

- 1) N and n are only reported as number of neurons (N) and number of boutons (n). The number of replicates (that is, separate experiments with different cultures) needs to be reported, and per-replicate data should also be shown somewhere to indicate the repeatability of the findings.
- 2) The sophisticated analysis pipeline is described in full detail. However, the code itself should also be provided.
- 3) Correct the typo "Sequience" in Fig. 3b.
- 4) Change the title to "asynchronous glutamate release..." to be consistent with the field.

Reviewer #2 (Remarks to the Author):

The paper of Mendonca et al investigates whether synchronously and asynchronously released vesicles constitutes the same pool in mouse cultured cortical glutamatergic neuron axon terminals, and whether they are released from the same subdomains of the active zone (AZ). They find negative correlation between release probability and the proportion of asynchronously released vesicles suggesting, that low release probability favors the asynchronous release mode. Using cutting edge techniques of detecting released glutamate with high localization precision (75 nm and 4 ms temporal resolution!) at many individual axon terminals synchronously they find, that asynchronous release sites are partially overlap with synchronous release sites, and synchronous release sites are more concentrated in the active zone, while the formers span on a wider range. They use a well characterized glutamate sensor (pAAV SF-iGluSnFR plasmid sparse transfection), fast camera acquisition, deconvolution and many image processing algorithms to distinguish between the 2 types of released vesicles, and the precise location of the release. They can also count the amount of vesicles released synchronously, and calculate the release probability from the deconvolved events. They suggest, that uneven voltage gated Ca²⁺ channel distribution in the AZ creates higher and lower release probability sites in the AZ, and sites further away from the Ca²⁺channel clusters are more prone to release asynchronously, while closer sites can release both synchronously and asynchronously with the action potentials. They provide detailed realistic computer modelling of vesicular release to support that hypothesis.

Two very recent publications have raised somewhat overlapping questions (Li et al 2021 Nat Commun., Kusick et al Nat Neurosci 2020), with different also cutting edge techniques, showing the relevance of the topic and also the technical challenges to address these questions. However, their conclusion is to some extent contradictory to the findings of this MS, as they propose, that asynchronous release is rather concentrated in the middle of the AZ rather than dispersed in its whole extent. As the authors use different experimental approach their data adds substantially to our concepts how the presynaptic release is organized.

The amount and the quality of the data obtained is sufficient and outstanding, the analyses is described well in the Methods, the very few missing parts are indicated bellow. Still I have few concerns both with the experimental design and the interpretation.

Major concerns:

1. All the imaging experiments were carried out at 23-25 oC instead of near physiological temperature, which changes many aspects of the release from Ca²⁺-channel kinetics, to diffusion, Ca²⁺-extrusion mechanisms, consequently the waveform of the Ca²⁺-transient, that is a key parameter of the release probability of the vesicles, and probably a very important parameter in the docking and priming of the vesicles and as such in the ratio of the synchronous and asynchronous release. Please at least discuss in the MS of the potential caveats of such a setting or provide some experimental data to show the differences / similarities and the potential changes it may cause in the interpretation.
2. What is the baseline frequency of spontaneous release from boutons? Does this frequency increase during the 5 Hz train of action potentials or there is an inherent baseline variability in the spontaneous release between boutons?
3. Estimation of total release (nT quantal / AP, and nA) at low release probability boutons is prone to high error if it is estimated from only 51 events. This can cause a substantial uncertainty in the estimation of the asynchronous release fraction (see large standard deviation of the data points on Panel A) potentially resulting in erroneously high values. Please estimate the error, and evaluate whether there is indeed a negative correlation between release probability (total release nT quantal / AP) and asynchronous release fraction taking into account the confidence intervals. I find it problematic to segregate a continuous population of data arbitrarily into two subpopulations at the e.g. median value and compare averages. Use of correlation analyses on the whole population would be more appropriate.
4. What is the reason the multi-quantal events were not analyzed in terms of localization? This is half of the total number of synchronous events. Can you exclude the possibility, that multivesicular release would occur at those sites that are seemingly only used for asynchronous release at the periphery of the AZ? If not, please state it in the MS.

Minor comments:

Methods

1. What is the rationale behind using voltage clamp and escape current to evoke release instead of evoking action potentials in current clamp mode? Do you have evidence, that escape current upon 5 ms depolarization to – 10 mV results in the same release properties or [Ca²⁺] transients as action potentials?
2. Modelling: Published Ca²⁺-channel density values are between 350 – 800 / μm^2 in cortical glutamatergic synapses for Cav2.1 and 2.2 (Kleindienst Int. J Mol. Sci. 2020 or Eltes et al J. Neurosci 2017). Calculating with the AZ area of 0.12 μm^2 used for the modelling suggests that Ca²⁺-channel

numbers are above 35 (at least 15 per cluster, if we assume 2 clusters in the AZ). How would it change the Ca²⁺ transient profile and the synchronous release probability map?

3. Modelling: What concentration of endogenous buffers (calmodulin, Calbindin D28K and ATP) were used for the simulations, and how does it relate to the intracellular solution used in the experiment, that contained 1 mM EGTA and Kgluconate both being Ca²⁺ buffers? (Woehler J. Phys. 2014). Implementing the buffers used in the intracellular solution changes any of the conclusion of the modelling?

4. What was the amount of the plasmid used for sparse expression of the SF-iGluSNFR?

Results

5. Please state clearly in the MS that the release probability used here is the probability of the release from the terminal and not the vesicle release probability. The former is the function of the size of the readily releasable pool and the vesicle release probability together.

Figures

Fig. 1. Panel C. I cannot recognize release event in any of the images in case of bouton No 2, however on Panel E it is shown to release basically at every action potential. What is the reason, that other boutons in a variable pattern display sign of release on the consecutive panels but not this selected one?

Fig. 2. Panel B. Asynchronous release fraction seems to saturate at value 1, what is the reason? If there is indeed a negative correlation with total release, at very low synchronous release cases the asynchronous release should exceed the synchronous, consequently the ratio would be larger than 1. Is there any technical reason for this plateau?

Fig. 2. Panel C(ii) same question as above for panel B, paired pulse ratio seems to plateau at 2.

Fig. 2. Panel C(iii) what is the percentage of boutons without asynchronous release? It seems that a very large percent of boutons independent of their total release does not display asynchronous release, please state is clearly in the MS.

Fig. 2. Panel C(iv) what could be the reason of an order of magnitude smaller asynchronous release while applying the paired pulse protocol?

Discussion

Please explain why only 2 Ca²⁺ channel clusters were implemented in the modelling of synchronous and asynchronous release, and why in the periphery of the AZ? None of the cited papers has any clear indication for such an arrangement. Neither Li et al 2021 Nat Comm nor Tang et al 2016 Nature show any Ca²⁺ channel localization, and in the very recent publication of Brockmann et al 2020 BioRxiv the $\alpha\delta$ subunit does not show any specific localization towards the periphery of the AZ. In Miki et al 2017 PNAS they correlate AZ area with Ca²⁺ channel cluster number in a glutamatergic terminal, and for an AZ size of 0.12 μm^2 the cluster number varies between 3-6. Other publications e.g. Kleindienst Int. J Mol. Sci. 2020 or Eltes et al J. Neurosci 2017 show Ca²⁺ channel distribution all over the AZ, and no preferential localization is visible. How would this change the conclusion of the modelling, if more Ca²⁺ channel clusters are implemented, and not necessary in the periphery of the AZ?

Reviewer #3 (Remarks to the Author):

Authors in this work developed a novel imaging technique to visualize the release of neurotransmitter vesicles from individual axon boutons. Fluorescent glutamate sensor SF-iGluSnFR was expressed on the axonal membrane of single cultured pyramidal cells, allowing the detection of glutamate release from individual synaptic vesicles with millisecond resolution. Using this method, the authors found that the ratio between synchronous and asynchronous synaptic vesicle exocytosis varies extensively among synapses formed by the same axon. Asynchronous release is enhanced at synapses with lower release probability and short-term facilitation. Further analysis revealed that, as compared with synchronous release sites, asynchronous release sites are more widely distributed across the presynaptic release area. Therefore, this manuscript provides new direct evidence showing heterogeneity of vesicle release mode at synapses formed by the same axon, and provides new insights into the balance between fast synchronous and delayed asynchronous release of neurotransmitters vesicles. I enjoyed reading the manuscript, the writing is very clear. The data is convincing, and the analysis is appropriate. Some minor concerns are listed below.

1. Considering the occurrence of baseline spontaneous release at many synapses, one may wonder whether there is any spontaneous vesicular release? It would be nice to show some traces with spontaneous release events. As asynchronous release and spontaneous release events are very difficult to be distinguished in some cases, especially when the cell is stimulated at low frequencies (5 or 20 Hz in the experiments), spontaneous release events (baseline without any stimulation and no action potential generation) should be included in the analysis.
2. The 10-ms time window for synchronous release is arbitrary. At room temperature and in cultured cells, the synaptic delay could be much longer. In previous studies, barrages of asynchronous release events could be detected after the cessation of a burst of high-frequency action potentials. If the data is available, it would be easier to distinguish the asynchronous release events from synchronous events.
3. Since the identity of postsynaptic cells for each axon boutons is unclear, it is difficult to link the findings to different types of synapses targeting to distinct types of cells.
3. The “total release nT” seems significantly higher in Syt1-KO mice as compared with WT mice (Fig2. A-B), suggesting that the presence of Syt1 in WT mice inhibits the amount of quanta per action potential. However, WT mice showed lower asynchronous release fraction, inconsistent with the conclusion that low release probability synapses hold higher asynchronous release fraction.
4. One of the “and that” should be removed in the abstract.
5. Some of the action potential markers were missing in Fig. 1E.
6. Is there any correlation between bouton size and release mode? see Fig. 3E.

7. In addition to the hierarchical cluster analysis shown in Fig. 3F, is it possible to examine the overlapping probability of release zone in different release modes?

8. The model work should be included in the results, not just in the discussion.

Response to the reviewers

Mendonca et al, Nature Communications Manuscript NCOMMS-21-19315-T

We are grateful to the reviewers for careful consideration of the manuscript and their valuable comments. We have taken on board all of their specific suggestions, and have substantially expanded the paper. The revised manuscript contains 2 new figures (Figs. 5 and 6) and 7 new supplementary figures (Supplementary Figs. 6 – 9 and 12 – 14). We believe that the revised manuscript is a much more definitive account of the differential tuning of synchronous and asynchronous release balance in synapses with different active zone morphologies.

The point-by-point response to each reviewer is provided below.

Reviewers' comments are italicised.

Our responses (as well as changes in the revised manuscript) are in blue colour.

For easy reference, queries from all three reviewers were renumbered throughout the response.

Please note that we cannot control the resolution of figures in the converted pdf file, therefore we refer the reviewers to the uploaded high-resolution figures if required.

Reviewer 1

Q1

N and n are only reported as number of neurons (N) and number of boutons (n). The number of replicates (that is, separate experiments with different cultures) needs to be reported, and per-replicate data should also be shown somewhere to indicate the repeatability of the findings.

We have followed the Reviewer's suggestion and included the information regarding the number of independent neuronal culture preparations in the figure legends. We also provide a source data file (*SourceData.xlsx*) which contains detailed information linking each recorded cell to the cell culture preparation. We used between 4 to 7 different cultures in each set of experiments presented in the main figures, and could not detect any differences in the presynaptic parameters measured among different culture preparations.

Q2

The sophisticated analysis pipeline is described in full detail. However, the code itself should also be provided.

We followed the reviewer's suggestion and submit a fully annotated software MATLAB and ImageJ codes together with the revised manuscript (*Mendonca_et_al_AnalysisScripts.zip*).

Q3

Correct the typo "Sequience" in Fig. 3b.

Thank you for pointing this out, the typo has been corrected.

Q4

Change the title to "asynchronous glutamate release..." to be consistent with the field.

Thank you for this suggestion. We have changed the title accordingly.

Reviewer 2

Major concerns:

Q5

All the imaging experiments were carried out at 23-25°C instead of near physiological temperature, which changes many aspects of the release from Ca²⁺ channel kinetics, to diffusion, Ca²⁺ extrusion mechanisms, consequently the waveform of the Ca²⁺ transient, that is a key parameter of the release probability of the vesicles, and probably a very important parameter in the docking and priming of the vesicles and as such in the ratio of the synchronous and asynchronous release. Please at least discuss in the MS of the potential caveats of such a setting or provide some experimental data to show the differences / similarities and the potential changes it may cause in the interpretation.

We agree with the reviewer that this is an important point. As suggested we discussed this in the revised manuscript as follows:

“...We note that our experiments were performed at ambient temperature (23–25 °C). It has been demonstrated that increasing the temperature to near physiological levels (32–35 °C) leads to an enhanced synchronisation of vesicular release and a concomitant decrease in the asynchronous release fraction (Huson et al., 2019; Pyott and Rosenmund, 2002). In large part, this is likely to be a consequence of temperature-dependent perturbation of the balance between the Ca²⁺ signal amplitude within Ca²⁺ microdomains in the active zone (which trigger synchronous release) and the global residual Ca²⁺ signal (which triggers asynchronous release). Indeed, an increase in temperature is expected to affect the major determinants of presynaptic Ca²⁺ dynamics, such as the action potential waveform, VGCC gating and Ca²⁺ buffering and extrusion. Nevertheless, we note that due to the differences in the stimulation patterns, our experiments shown in Fig. 2 and Fig. 3 represent two contrasting cases with different ratios between the local Ca²⁺ signal in the active zone and the global residual Ca²⁺ signal. During repetitive stimulation (Figs. 1 and 2, 5 Hz train) the global Ca²⁺ signal is expected to build up several fold in comparison to that after a pair of action potentials (Fig. 3) (Chamberland et al., 2020). In line with this, the asynchronous release fraction was ~3.5-fold higher during 5 Hz trains ($n_A/n_T \sim 0.25$) than during paired-pulse recordings ($n_A/n_T \sim 0.07$). Still, in both cases, we observed an inverse relationship between n_A/n_T and the overall release rate, n_T . This result therefore argues that the same relationship between release rate and n_A/n_T is likely to be observed at physiological temperatures, despite the changes in presynaptic Ca²⁺ dynamics and the overall synchronisation of release. ...”

Q6

What is the baseline frequency of spontaneous release from boutons? Does this frequency increase during the 5 Hz train of action potentials or there is an inherent baseline variability in the spontaneous release between boutons?

We followed the Reviewers' 2 and 3 suggestions and performed control experiments to estimate the baseline frequency and the contribution of spontaneous release to our measurements of the asynchronous release component. These data are included in the revised manuscript (main text and Supplementary Fig. 6).

In summary, spontaneous release in the absence of action potentials (when the cell was held at -70 mV) occurred very rarely in comparison to asynchronous release during 5 Hz stimulation. During the 10-second imaging window (corresponding to the duration of 5 Hz train of 51 action potentials), we detected spontaneous events only in 13.3 ± 3.0 % of boutons. In contrast, we observed asynchronous events in 85.6 ± 1.7 % of boutons (Supplementary Fig. 6E). Furthermore, the average fraction of spontaneous release normalised to the total evoked release was ~ 0.8 . This value is ~ 30 times lower than the average asynchronous release fraction during 5 Hz stimulation: 25 % (Supplementary Fig. 6F and Fig.1G). Thus, we concluded that spontaneous release has a negligible contribution to the estimates of evoked release in our conditions.

Q7

Estimation of total release (n_T quantal / AP, and n_A) at low release probability boutons is prone to high error if it is estimated from only 51 events. This can cause a substantial uncertainty in the estimation of the asynchronous release fraction (see large standard deviation of the data points on Panel A) potentially resulting in erroneously high values. Please estimate the error, and evaluate whether there is indeed a negative correlation between release probability (total release n_T quantal / AP) and asynchronous release fraction taking into account the confidence intervals.

I find it problematic to segregate a continuous population of data arbitrarily into two subpopulations at the e.g. median value and compare averages. Use of correlation analyses on the whole population would be more appropriate.

We thank the reviewer for bringing our attention to this important point. To address this, we implemented the following additional analysis:

- (1) We acknowledged the problem with estimating n_A/n_T and n_T in low release probability synapses during a single trial and modified text in the Results section as follows:

“...estimates of n_A/n_T and n_T are prone to high error if only one trial is used. One way to overcome this difficulty was to record several sweeps from the same ROI and calculate the average n_A/n_T . However, due to the limitations caused by phototoxicity, this would restrict our recordings to a single ROI and would also significantly reduce the number of recorded boutons and the overall experimental success rate. We therefore chose an alternative approach based on grouping boutons according to n_T and calculating the average n_A/n_T in each group...”

- (2) As suggested, we performed the estimates of confidence intervals for both n_T and n_A/n_T in boutons with low release rates (new Supplementary Fig.8). This analysis allowed us to optimally group the boutons in each cell for averaging – in three groups

of equal size (terciles) with low, intermediate and high n_T values. In this way we demonstrated that the observed increase of n_A / n_T was specific to the low release rate boutons (with $n_T < 0.2$) (new Fig. 2 and Fig. 3 and the updated text in the Results section).

- (3) As suggested by the reviewer, we included correlation analysis on the whole bouton populations in individual cells (Supplementary Fig. 7D), which further confirmed our findings.
- (4) Finally, we showed that the increase in the asynchronous release fraction in synapses with low n_T was not reproduced in control Monte-Carlo simulations, where we randomly reassigned the type of each recorded release event (synchronous or asynchronous) in each bouton independently of n_T using the cell-averaged n_A / n_T value (Supplementary Fig. 9). This argues that the observed phenomenon was not simply a consequence of a spurious correlation.

Q8

What is the reason the multi-quantal events were not analysed in terms of localisation? This is half of the total number of synchronous events. Can you exclude the possibility, that multivesicular release would occur at those sites that are seemingly only used for asynchronous release at the periphery of the AZ? If not, please state it in the MS.

The sub-pixel event localisation analysis was based on 2D gaussian fitting of iGluSnFR spatial profiles by adapting the algorithms that are used in single-molecule localisation microscopy, which were specifically designed to estimate the localisation of events known to have a single local maximum (*i.e.*, single quanta events). We did try to adapt this analysis for fitting multi-quantal events, however due to limited signal-to-noise ratio in most cases the algorithm was unable to distinguish two local maxima, returning the average location of both events. We note that the analysis of the relative distributions of multivesicular and univesicular release events was already performed with vGluT1-pHluorin imaging (which has ~3 fold better spatial resolution than SF-iGluSnFR imaging)(Maschi and Klyachko, 2020) and revealed that multivesicular release events are preferentially located towards the centre and not to the periphery of the release area. This argues against the possibility that multivesicular release would occur at those sites that are seemingly only used for asynchronous release. As suggested by the reviewer we have included these considerations in the manuscript discussion.

Minor comments:

Methods

Q9

What is the rationale behind using voltage clamp and escape current to evoke release instead of evoking action potentials in current clamp mode? Do you have evidence, that escape current upon 5 ms depolarisation to – 10 mV results in the same release properties or [Ca²⁺] transients as action potentials?

We agree with the reviewer that using current-clamp mode seems at first sight a more suitable option to evoke action potentials in a more physiological way. In fact, this was the approach used in our previous work (Tagliatti et al., PNAS 2020). However, in the current-clamp mode we could not control any firing adaptation the cell might display over the course of stimulation (e.g. burst firing or changes in excitability caused by K^+ and Ca^{2+} conductances). Indeed, we observed missing spikes, double action potentials or multiple spikes during the 5Hz x 51APs trains (Reply to Reviewers Figure 1). Therefore, such trials had to be excluded during the analysis.

In contrast, the use of 5 ms voltage steps from -70 mV to -10 mV allowed us to reliably evoke individual action potentials (as long as the series resistance was within an acceptable range of $\sim 20M\Omega$). We note that our imaging ROIs were typically located between $\sim 100 - 300 \mu m$ from the soma. Therefore, it is likely that at such long distance the action potential waveform is not significantly affected by the brief somatic depolarisation and is predominantly determined by the intrinsic conductance of the axon (e.g. see Fig. 2 in Scott et al., J. Neurosci 2008).

In line with the above considerations, we found that the rates of both synchronous and asynchronous release in wild type neurons during 5 Hz stimulation reported in this manuscript (recorded in voltage-clamp, Fig. 1G) were similar to those reported in our previous work where we used the current-clamp protocol (Fig 3B, C in Tagliatti et al., PNAS 2020).

Modelling: Published Ca²⁺channel density values are between 350 – 800 / μm^2 in cortical glutamatergic synapses for Cav2.1 and 2.2 (Kleindienst Int. J Mol. Sci. 2020 or Eltes et al J. Neurosci 2017). Calculating with the AZ area of 0.12 μm^2 used for the modelling suggests that Ca²⁺channel numbers are above 35 (at least 15 per cluster, if we assume 2 clusters in the AZ). How would it change the Ca²⁺ transient profile and the synchronous release probability map?

We followed the Reviewer's suggestion and expanded the computational modelling section:

- (1) In the revised manuscript, we systematically explored the relationship between the active zone structural organisation and the spatial distribution of synchronous and asynchronous release (new Figs. 5 and 6, and Supplementary Figs. 12, 13 and 14). We considered nine distinct cases of VGCC channel distribution in the active zone that are illustrated in Fig. 5B. We varied the number of VGCC clusters (1 – 5 range) and also their relative distribution within the active zone. VGCC clusters were either located more to the centre (Cases i, ii, iv, v, vii and ix) or to the periphery (Cases iii, vi and viii) of the active zone. In line with the experimental estimates, the overall channel density for all considered cases was $\sim 500 \mu\text{m}^{-2}$.
- (2) We calculated synchronous release probability maps for each of the considered geometries (Fig. 5B, C) and performed simulations of vesicular release for different (random) distributions of RRP vesicles.
- (3) The extended model confirmed our initial finding that heterogeneity of coupling distance between RRP vesicles and VGCCs could partly account for the elevated asynchronous release fraction in low release rate synapses (old Fig. 4 and new Figs 5 and 6 and Supplementary Fig 14).
- (4) Furthermore, the extended model revealed several general principles that demonstrate how the active zone morphology determines the rates and distributions of synchronous and asynchronous release. The model predicts that the average probability of synchronous release of individual RPP vesicles (p_s) increases with the size of the active zone (Fig. 6C). This happens because, on average, larger active zones contain more VGCC clusters with overlapping Ca²⁺ microdomains. Due to the high Ca²⁺ cooperativity of synchronous release, this Ca²⁺ microdomain overlap leads to a significant increase of p_s for RRP vesicles that are located in the vicinity of 2 or more VGCC clusters (Fig. 5C). In contrast, asynchronous release is triggered by the global ('residual') increase in the presynaptic Ca²⁺ concentration after the collapse of transient nano/micro domains (Fig. 5A). Therefore, the probability of asynchronous release (p_a) is not expected to depend on the distribution of VGCCs in the active zone (Fig. 6D).
- (5) The model reproduced both a greater asynchronous release fraction at low release rate boutons and a wider distribution of asynchronous release loci within the release area. This argues that the natural variability in the active zone structural organisation (including active zone size, VGCC cluster number and their relative distribution with respect to RRP vesicles) is the major factor that regulates the balance between synchronous and asynchronous release among synaptic outputs of pyramidal cells.

Q11

Modelling: What concentration of endogenous buffers (calmodulin, Calbindin D28K and ATP) were used for the simulations, and how does it relate to the intracellular solution used in the experiment, that contained 1 mM EGTA and Kgluconate both being Ca²⁺-buffers? (Woehler J. Phys. 2014). Implementing the buffers used in the intracellular solution changes any of the conclusion of the modelling?

As suggested by the reviewer, in the revised manuscript we considered both limiting cases: a presynaptic bouton with endogenous buffers (calmodulin, Calbindin D28k and ATP) and a bouton filled with the intracellular pipette buffer (EGTA, ATP, K⁺-gluconate). The kinetics reactions and bindings rates for the endogenous and exogenous buffers are now provided in Supplementary Tables 1 and 2 respectively.

The outputs were very similar for both models, and the model with the intracellular pipette buffer reproduced all the conclusions obtained in the model with the endogenous buffers (see Fig. 5 and Supplementary Fig. 12; Fig. 6 and Supplementary Fig. 13, and Supplementary Fig. 14 A and B).

Q12

What was the amount of the plasmid used for sparse expression of the SF-iGluSNFR?

450ng per coverslip. This information has been added to the Materials and Methods section.

Results

Q13

Please state clearly in the MS that the release probability used here is the probability of the release from the terminal and not the vesicle release probability. The former is the function of the size of the readily releasable pool and the vesicle release probability together.

We agree with the reviewer that it was important to clarify this. We modified text in the “Imaging of synchronous and asynchronous vesicular release across axonal arbour” results section as follows:

“...For each bouton, we calculated the total release rate, n_T , by dividing the sum of all quanta released during the train by the number of action potentials in the train ($N_{AP} = 51$). We note that n_T is directly related to the overall release probability, P_{rel} . Indeed, let us consider a synapse with RRP size of m vesicles and with an average release probability of individual RRP vesicles p_v . According to binomial statistics, the probability that at least one vesicle is released at a given action potential is $P_{rel} = 1 - (1 - p_v)^m$ and the total release rate (defined as the average number of vesicles released per action potential) is $n_T = mp_v$. In low release probability synapses, $n_T \approx P_{rel}$. However, in synapses that release on average more than one vesicle per action potential, the use of P_{rel} is not optimal as it saturates at 1 (e.g. Bouton 2 in Fig. 1E). By contrast, n_T provides a linear readout of vesicular release

rate in all types of synapses. Therefore, we used n_T in subsequent analysis. In full agreement with previous studies (e.g. (Ermolyuk et al., 2012; Jensen et al., 2019)) we found that n_T varied widely among presynaptic boutons supplied by the same axon (~0.01 – 2.2 range, Supplementary Fig. 4 and also Figs. 2 and 3 below)…”

Figures

Q14

Fig. 1. Panel C. I cannot recognise release event in any of the images in case of bouton No 2, however on Panel E it is shown to release basically at every action potential. What is the reason, that other boutons in a variable pattern display sign of release on the consecutive panels but not this selected one?

We apologise for the misinterpretation, which was due to the low-resolution traces in the PDF file sent to reviewers. Bouton 2 does indeed fail at the 1st AP, and coincidentally display single quanta responses (low ΔF) at the 21st and 41st action potential. However, the dashed line indicating the 1st AP was lost in the compressed PDF, causing the impression that there was a substantial SF-iGluSnFR response at the 1st AP, when in fact it occurred at the 2nd AP (Reply to Reviewers Figure 2). We have rectified this problem by thickening the lines that depict action potential timings. We also included the image-stacks containing this specific bouton as an example data set along the executable analysis scripts (Mendonca_et_al_AnalysisScripts.zip, included in the revised submission), so that the reviewer can confirm its release profile if convenient.

Q 15
Q 16

Fig. 2. Panel B. Asynchronous release fraction seems to saturate at value 1, what is the reason? If there is indeed a negative correlation with total release, at very low synchronous release cases the asynchronous release should exceed the synchronous, consequently the ratio would be larger than 1. Is there any technical reason for this plateau?

Fig. 2. Panel C(ii) same question as above for panel B, paired pulse ratio seems to plateau at 2.

Yes, please note that to avoid division by zero, the asynchronous release fraction was defined as the ratio of asynchronous release rate to total release rate: $n_A / n_T = n_A / (n_S + n_A)$. Therefore, this value can only vary from 0 to 1: an asynchronous release fraction = 1 means that all quanta were released asynchronously, and an asynchronous release fraction = 0 means that all quanta were released synchronously. We have modified text in the section 'Asynchronous release is elevated in synapses with low release rate' to clarify this.

Similarly, the classical formula for paired-pulse ratio used in electrophysiological recordings ($PPR = N_2 / N_1$) was adapted to avoid division by zero and the paired-pulse ratio in individual boutons was calculated as $PPR = 2N_2 / (N_1 + N_2)$, where N_1 and N_2 are total numbers of quanta released at first and second action potentials respectively. If during the paired pulse protocol a given synapse displays release only at the first action potentials (*i.e.* $N_2 = 0$, maximal condition for short term depression), then $PPR = 0$. Conversely, if release occurs only at the second action potentials (*i.e.* $N_1 = 0$, maximum condition for short term facilitation), then $PPR = 2$. If $N_1 = N_2$ (no facilitation or depression), then $PPR = 1$. Note, that the first version of our manuscript had a typo in the PPR formula (missing the factor 2), we have corrected this and apologise for the mistake.

We have modified text in the section 'Relationship between short-term facilitation and synchronous/asynchronous release balance' to clarify this.

Q17

Fig. 2. Panel C(iii) what is the percentage of boutons without asynchronous release? It seems that a very large percent of boutons independent of their total release does not display asynchronous release, please state is clearly in the MS.

The appearance of boutons with only synchronous release events is a consequence of limited sampling. In the revised version of the manuscript we discussed this in the preceding results section "Asynchronous release is elevated in synapses with low release rate" as follows:

"...In large part, this variability was due to limited sampling (especially in boutons with a low release rate). For example, let us consider a bouton that on average releases only 2 vesicles during the stimulus train and has an expected value of $n_A / n_T = 0.5$. Then, during a single trial, there is a 25% chance of observing 2 synchronous events, a 25 % chance of observing 2 asynchronous events and a 50% chance of observing 1 synchronous and 1 asynchronous event..."

In the case of results presented in new Fig. 3 (old Fig.2C) due to the low values of $n_A / n_T \sim 0.07$ and because only 20 action potentials were recorded per trial, the same limitation also applies to high release rate synapses. Indeed, let us consider a bouton with $n_T = 1$, which releases on average 20 vesicles during 10 paired pulse trials. Then it follows that the probability that a given release event was synchronous is $(1-0.07) = 0.93$, and that all 20 vesicles underwent synchronous release is $0.93^{20}=0.23$, which is reflected in the figure. We note that the estimates of confidence intervals in Supplementary Fig. 8 were performed for a 51 action potential train. In the case of paired pulse protocol, we sampled each bouton with only 20 action potentials. Therefore, to stay within the same confidence intervals, in the case of paired pulse analysis we needed to average at least 17 boutons per group (tercile). We therefore only included cells that contained at least 50 boutons. We also increased the overall number of recorded cells from 13 to 20.

Q18

Fig. 2. Panel C(iv) what could be the reason of an order of magnitude smaller asynchronous release while applying the paired pulse protocol?

As we discussed in the manuscript and in our answer to Q5 above:

"...During repetitive stimulation (Figs. 1 and 2, 5 Hz train) the global Ca^{2+} signal is expected to build up several fold in comparison to that after a pair of action potentials (Fig. 3)(Chamberland et al., 2020). In line with this, the asynchronous release fraction was ~ 3.5 -fold higher during 5 Hz trains ($n_A / n_T \sim 0.25$) than during paired-pulse recordings ($n_A / n_T \sim 0.07$)..."

Indeed, Fig. 1G shows that during 5 Hz stimulation, asynchronous release rate n_A in WT cells nearly doubles from the initial value of 0.07 at the first action potential, to 0.13 during the steady state part of the train (action potentials 20 - 51), while synchronous release rate n_S decreases by a third due to short-term depression (from ~ 0.9 to ~ 0.36 quanta/AP), which corresponds to the increase of asynchronous release fraction n_A / n_T from 0.07 to 0.27.

We also note that in our initial submission the average n_A / n_T estimated during paired pulse protocol was lower (~ 0.05 , old Fig. 2C) than that in the revised version (~ 0.07 , Fig. 3). This was due to the bias that we introduced by selecting boutons only on the basis of synchronous release in the first submission. We have corrected this, and we thank the reviewer for pointing out to this issue.

Discussion

Q19

Please explain why only 2 Ca²⁺channel clusters were implemented in the modelling of synchronous and asynchronous release, and why in the periphery of the AZ? None of the cited papers has any clear indication for such an arrangement. Neither Li et al 2021 Nat Comm nor Tang et al 2016 Nature show any Ca²⁺channel localisation, and in the very recent publication of Brockmann et al 2020 BioRxiv the $\alpha 2\delta$ subunit does not show any specific localisation

towards the periphery of the AZ. In Miki et al 2017 PNAS they correlate AZ area with Ca²⁺channel cluster number in a glutamatergic terminal, and for an AZ size of 0.12 μm² the cluster number varies between 3-6. Other publications e.g. Kleindienst Int. J Mol. Sci. 2020 or Eltes et al J. Neurosci 2017 show Ca²⁺channel distribution all over the AZ, and no preferential localisation is visible. How would this change the conclusion of the modelling, if more Ca²⁺channel clusters are implemented, and not necessary in the periphery of the AZ?

As stated above, we have now considered nine different cases with different numbers and distributions of VGCC within the active zone with both more central and more peripheral locations (please see our detailed response to Q10 above).

We note that our initial model with the peripheral location of VGCCs in the active zone was based on the following considerations: (i) Brockmann et al 2020 BioRxiv demonstrated that RRP vesicles preferentially align with Ca²⁺-channels and postsynaptic AMPA receptors within 20-30 nm. (ii) At the same time, Li et al 2021 Nat Comm showed that AMPA receptor clusters are preferentially located to the periphery of the active zone. (iii) Then, it follows that VGCC could also be preferentially located to the periphery of the active zone.

However, we agree with the reviewer that the available data from EM analysis of VGCCs do not support preferential location VGCC to the active zone periphery and acknowledge this in the manuscript

Reviewer 3

Q20

Considering the occurrence of baseline spontaneous release at many synapses, one may wonder whether there is any spontaneous vesicular release? It would be nice to show some traces with spontaneous release events. As asynchronous release and spontaneous release events are very difficult to be distinguished in some cases, especially when the cell is stimulated at low frequencies (5 or 20 Hz in the experiments), spontaneous release events (baseline without any stimulation and no action potential generation) should be included in the analysis.

We followed the Reviewers' 2 and 3 suggestions and performed control experiments to estimate the baseline frequency and the contribution of spontaneous release to our measurements of the asynchronous release component. These data are included in the revised manuscript (main text and Supplementary Fig. 6).

In summary, spontaneous release in the absence of action potentials (when the cell was held at -70 mV) occurred very rarely in comparison to asynchronous release during 5 Hz stimulation. During the 10-second imaging window (corresponding to the duration of 5 Hz train of 51 action potentials), we detected spontaneous events only in 13.3 ± 3.0 % of boutons. In contrast, we observed asynchronous events in 85.6 ± 1.7 % of boutons (Supplementary Fig. 6E). Furthermore, the average fraction of spontaneous release normalised to the total evoked release was ~ 0.8 . This value is ~ 30 times lower than the average asynchronous release fraction during 5 Hz stimulation: 25 % (Supplementary Fig. 6F and Fig.1G). Thus, we concluded that spontaneous release has a negligible contribution to the estimates of evoked release in our conditions.

Q21

The 10-ms time window for synchronous release is arbitrary. At room temperature and in cultured cells, the synaptic delay could be much longer. In previous studies, barrages of asynchronous release events could be detected after the cessation of a burst of high-frequency action potentials. If the data is available, it would be easier to distinguish the asynchronous release events from synchronous events.

We agree with the reviewer that it was important to verify that 10 ms is indeed an optimal threshold to separate synchronous and asynchronous release. In fact, when choosing this threshold, we did consider our control experiments with paired cell recordings in our preparation at room temperature (Reply to Reviewers Figure 3). We found that the latency of synaptic responses was typically within a range of 2 – 8 ms. We also performed experiments using minimal axonal stimulation (monopolar electrodes) in the same conditions and obtained similar values for the latency between stimulation and the peak of the AMPA receptor-mediated post-synaptic response – range: 3 – 9 ms; mean \pm SD: 5.3 ± 2.2 ms N = 6 cells from 3 independent preparations).

We agree with the reviewer that there are other methods suited to evoke asynchronous release, which include measuring release after high frequency firing. However, in this study, we sought to obtain the spatiotemporal profile of both synchronous and asynchronous release from the same stimulation protocol (most data shown contrast both release modes, as seen in Figs. 2, 3 and 4).

Q22

Since the identity of postsynaptic cells for each axon boutons is unclear, it is difficult to link the findings to different types of synapses targeting to distinct types of cells.

We agree that identifying the type of postsynaptic target cells would be an interesting and most natural forward step in this project. We are currently addressing this issue by performing post-hoc immunofluorescence staining of the recorded iGluSnFR-expressing neurons in order to correlate the activity on the presynaptic site to the identity of postsynaptic cell. However, we hope that the reviewer will agree that this lies outside the scope of the current manuscript, We have also emphasised in the discussion that this hypothesis requires further testing in different types of synapses.

Q 23

The “total release n_T ” seems significantly higher in Syt1-KO mice as compared with WT mice (Fig2. A-B), suggesting that the presence of Syt1 in WT mice inhibits the amount of quanta per action potential. However, WT mice showed lower asynchronous release fraction, inconsistent with the conclusion that low release probability synapses hold higher asynchronous release fraction.

Although wild type and Syt1^{-/-} synapses differ significantly in the amount of synchronous and asynchronous vesicle release, it cannot be concluded that the total release rate n_T is higher in Syt1-KO neurons (Reply to Reviewers Figure 4, Fig 1G in the manuscript). The left plots from Fig. 2A and B, which are mentioned by the reviewer, are from representative wild type and Syt1^{-/-} cells, where each datapoint is a bouton, and therefore insufficient to infer any difference for total release between genotypes. Nevertheless, Fig. 1G (bottom panel) shows

that total quantal release rates during the steady-state part of the train averaged among all recorded cells are similar in the two genotypes. To clarify this point, we added a proper quantification on different modes of release in wild type and *Syt1*^{-/-} neurons to the revised Fig. 1G (right panels).

Q24

One of the “and that” should be removed in the abstract.

Thank you for pointing this out. This typo has been corrected.

Q25

Some of the action potential markers were missing in Fig. 1E.

We apologise for this issue. This was due to the decrease in resolution in the automatically generated PDF document that was sent to the reviewers. We have rectified this issue by thickening all lines in Figure 1.

Q26

Is there any correlation between bouton size and release mode? see Fig. 3E.

We did not find any significant correlation between the release area (old Fig. 3E, new Fig. 4E) and the mode of release among analysed boutons in the present analysis. It is important to highlight that a significant increase of asynchronous release fraction was only observed in one third of boutons with the lowest release rate (Fig.2A and Supplementary Fig. 7): tercile T1, average $n_T=0.11$, range 0.04 – 0.25. On the other hand, due to technical limitations (please see Methods, Data Inclusion and Exclusion Criteria), the spatial localisation data shown in Fig.4E only include boutons with $n_T \geq 0.1$.

Q27

In addition to the hierarchical cluster analysis shown in Fig. 3F, is it possible to examine the overlapping probability of release zone in different release modes?

We indeed tried to implement such an analysis. However, it was difficult to interpret the results because the overlapping probability of release zones for different release modes (i.e. overlapping area normalised to the total release area) depended on the relative numbers of fitted synchronous and asynchronous events that varied due to limited sample size and exclusion of multiquantal events.

Q28

The model work should be included in the results, not just in the discussion.

As suggested, we have moved modelling to the results section

REVIEWERS' COMMENTS

Reviewer #1 (Remarks to the Author):

All my points have been addressed.

I just have one note on interpretation, which I failed to realize during the first round of review. The authors found a link between PPR, synchronicity, and release probability. It has been proposed (Miki et al., 2018) that a mechanism for both facilitation and asynchronous release is the rapid (ms to 10s of ms) activity-dependent transient recruitment of docked synaptic vesicles (Miki et al., 2016; Kusick et al., 2020). In this scheme, per action potential an initially undocked vesicle cannot fuse synchronously, but it can dock and either (with low probability) fuse right away in 'two-step' release, which will by definition be asynchronous, or dock and remain ready for the next action potential, thereby contributing to facilitation/reducing depression. Therefore, one difference among synapses that could help account for the authors findings is the difference in ratio of docked to undocked vesicles and t . Such a relationship has been proposed to account for differences between different synapse types (Maus et al., 2020, Neher and Brose, 2018). Note that this doesn't mean all asynchronous release is via two-step release, so is more a possibility in addition to, rather than instead of, the authors' spatial model.

This need not be presented as an alternative to the author's very thorough spatial model of the active zone, which can account for their findings (there is nothing mutually exclusive about these two explanations, I would guess that both these 'horizontal' and 'vertical' spatial distributions of vesicles are important, and only the author's proposed scheme can account for the difference in locations of synch and asynch, since docked vesicles are evenly distributed across the active zone). However, since it is a proposal in the field consistent with the relationship between release probability, synchronicity, and short-term plasticity that the authors found, it should be briefly mentioned. If they don't wish to get into this whole docking idea, they should at least cite Miki et al 2018 as simply a paper that proposed a link between short-term plasticity and asynchronous release.

Reviewer #2 (Remarks to the Author):

The authors replied all my comments. I have no further questions or concerns.

Response to the reviewers

Reviewer 1

I just have one note on interpretation, which I failed to realize during the first round of review. The authors found a link between PPR, synchronicity, and release probability. It has been proposed (Miki et al., 2018) that a mechanism for both facilitation and asynchronous release is the rapid (ms to 10s of ms) activity-dependent transient recruitment of docked synaptic vesicles (Miki et al., 2016; Kusick et al., 2020). In this scheme, per action potential an initially undocked vesicle cannot fuse synchronously, but it can dock and either (with low probability) fuse right away in 'two-step' release, which will by definition be asynchronous, or dock and remain ready for the next action potential, thereby contributing to facilitation/reducing depression. Therefore, one difference among synapses that could help account for the authors findings is the difference in ratio of docked to undocked vesicles and t. Such a relationship has been proposed to account for differences between different synapse types (Maus et al., 2020, Neher and Brose, 2018). Note that this doesn't mean all asynchronous release is via two-step release, so is more a possibility in addition to, rather than instead of, the authors' spatial model.

This need not be presented as an alternative to the author's very thorough spatial model of the active zone, which can account for their findings (there is nothing mutually exclusive about these two explanations, I would guess that both these 'horizontal' and 'vertical' spatial distributions of vesicles are important, and only the author's proposed scheme can account for the difference in locations of synch and asynch, since docked vesicles are evenly distributed across the active zone). However, since it is a proposal in the field consistent with the relationship between release probability, synchronicity, and short-term plasticity that the authors found, it should be briefly mentioned. If they don't wish to get into this whole docking idea, they should at least cite Miki et al 2018 as simply a paper that proposed a link between short-term plasticity and asynchronous release.

We thank the reviewer for this suggestion. We have added the following text to the discussion:

'...Several recent studies have suggested that activity-dependent multi-step regulation of transient synaptic vesicle recruitment at the active zone is a major factor in determining efficacy, kinetics and plasticity of synaptic vesicle release at different types of synapses(Chang et al., 2018;Kusick et al., 2020;Maus et al., 2020;Miki et al., 2018;Neher and Brose, 2018). It remains to be established whether vesicular docking and priming states (and as a consequence, release site occupancy) are differentially regulated among presynaptic boutons supplied by the same axon. If this is indeed the case, this mechanism should also contribute to the interdependent regulation of vesicular release kinetics and short-term synaptic plasticity at a single bouton level....'

Reference List

Chang,S., Trimbuch,T., and Rosenmund,C. (2018). Synaptotagmin-1 drives synchronous Ca(2+)-triggered fusion by C2B-domain-mediated synaptic-vesicle-membrane attachment. Nat. Neurosci. 21, 33-40.

Kusick,G.F., Chin,M., Raychaudhuri,S., Lippmann,K., Adula,K.P., Hujber,E.J., Vu,T., Davis,M.W., Jorgensen,E.M., and Watanabe,S. (2020). Synaptic vesicles transiently dock to refill release sites. *Nat. Neurosci.* 23, 1329-1338.

Maus,L., Lee,C., Altas,B., Sertel,S.M., Weyand,K., Rizzoli,S.O., Rhee,J., Brose,N., Imig,C., and Cooper,B.H. (2020). Ultrastructural Correlates of Presynaptic Functional Heterogeneity in Hippocampal Synapses. *Cell Rep.* 30, 3632-3643.

Miki,T., Nakamura,Y., Malagon,G., Neher,E., and Marty,A. (2018). Two-component latency distributions indicate two-step vesicular release at simple glutamatergic synapses. *Nat. Commun.* 9, 3943-06336.

Neher,E., and Brose,N. (2018). Dynamically Primed Synaptic Vesicle States: Key to Understand Synaptic Short-Term Plasticity. *Neuron.* %19;100, 1283-1291.